# mFABIO: An integrative multi-tissue TWAS fine-mapping approach to prioritize potentially causal genes and tissues underlying binary traits

**Haihan Zhang**[1], **Kevin He**[1], **Lam C. Tsoi**[1,2,3], **Xiang Zhou**[4]*

**1** Department of Biostatistics, University of Michigan, Ann Arbor, Michigan, United States of America, **2** Department of Dermatology, University of Michigan Michigan Medicine, Ann Arbor, Michigan, United States of America, **3** Department of Computational Medicine and Bioinformatics, University of Michigan, Ann Arbor, Michigan, United States of America, **4** Department of Statistics and Data Science, Yale University, New Haven, Connecticut, United States of America

* xiang.zhou.xz735@yale.edu

## Abstract

Recent advances in transcriptome-wide association study (TWAS) fine-mapping have enabled the joint modeling of multiple genes to improve causal gene prioritization. However, existing methods have been developed primarily for quantitative traits and most of them rely on gene expression data from a single tissue. Here, we present mFABIO, a multi-tissue TWAS fine-mapping method specifically designed for binary traits. mFABIO employs a probit model to directly link genetically regulated expression (GReX) of genes within a locus across multiple tissues to a binary outcome, while accounting for correlations in GReX across genes and tissues. As a result, mFABIO offers substantial power gains for binary traits, while maintaining robust control of false discovery rates (FDR). We evaluated mFABIO through extensive simulations and applied it to an in-depth analysis of six binary disease traits (asthma, breast cancer, gout, hypertension, prostate cancer, and rheumatoid arthritis) in the UK Biobank, using expression data spanning 38 Genotype-Tissue Expression (GTEx) tissues. mFABIO identified an average of 42 likely causal genes and 65 tissue-gene pairs per disease (FDR < 0.05). Notably, 60.9% of the genes and 77.2% of the gene-tissue pairs were supported by existing TWAS or GWAS evidence. This represented at least a 14.9% increase in evidence-supported genes and a 14.8% increase in evidence-supported gene-tissue pairs, compared to existing approaches. Additionally, mFABIO was also able to narrow down the list of potentially causal candidates by at least 51.3% for genes, and 50.8% for gene-tissue pairs, compared to single-tissue approaches. Leveraging its improved power, mFABIO successfully prioritized multiple potentially causal gene-tissue pairs associated with these diseases, with biological support. Notable examples include *D2HGDH* in lung tissue for asthma, *CYBRD1* in breast mammary tissue for breast cancer, and *CCR6* in spleen tissue

**Data availability statement:** The genetically regulated expression (GReX) prediction models we used in this study are publicly available at https://doi.org/10.7910/DVN/8IPOPK. The method mFABIO we introduced in this study is implemented as an R package, freely available at https://superggbond.github.io/mFABIO/.

**Funding:** L.C.T. receives two grants from National Institutes of Health (R01AR080662; UC2 AR081033). X.Z. receives two grants from National Institutes of Health (R01HG009124; R01GM144960). The funders had no role in study design, data collection and analysis, decision to publish, or preparation of the manuscript.

**Competing interests:** I have read the journal's policy and the authors of this manuscript have the following competing interests: L.C.T. has received support from Galderma and Janssen. All other authors have declared that no competing interests exist.

for rheumatoid arthritis. Overall, mFABIO serves as an effective tool for multi-tissue TWAS fine-mapping of binary traits.

## Author summary

Many genetic variants influence disease risk by altering how genes are expressed in different tissues. Transcriptome-wide association studies (TWAS) help identify such genes, but most existing methods analyze one tissue at a time and are designed for quantitative traits rather than diseases with binary outcomes. We developed mFABIO, a new method that improves the identification of likely causal genes for binary disease traits by jointly modeling gene expression across multiple tissues. By accounting for correlations between genes and tissues, mFABIO increases the power to detect disease-related genes while controlling false discoveries. Using UK Biobank genetic data and gene expression from 38 tissues in the GTEx project, we applied mFABIO to six diseases: asthma, breast cancer, gout, hypertension, prostate cancer, and rheumatoid arthritis. mFABIO identified more biologically supported genes than existing methods while substantially narrowing the list of candidate genes. These results demonstrate that mFABIO can improve the discovery and prioritization of genes and tissues involved in complex diseases.

## Introduction

Transcriptome-wide association studies (TWAS) integrate gene expression data with genome-wide association studies (GWAS) to identify potentially causal gene-trait associations [1–3]. Conventional TWAS methods test the marginal association between the genetically regulated expression (GReX) of a single gene and a trait through a two-step process [1,2,4]. First, expression prediction models are built using local (cis-) single nucleotide polymorphisms (SNPs) in a gene expression dataset. Second, these models are used to impute GReX in the GWAS cohort and test its association with the trait of interest. By aggregating SNP-level signals into a single gene-based test, TWAS can increase statistical power and provide mechanistic insights into disease biology. However, a key limitation of these marginal approaches is confounding by nearby genes, as neighboring genes often exhibit correlated expression and are regulated by shared cis-SNPs in high linkage disequilibrium (LD) [4,5]. Consequently, a gene identified by marginal TWAS may simply be a non-causal "tag gene" rather than the true causal one.

To address the confounding inherent in marginal analyses, TWAS fine-mapping methods were developed to prioritize causal genes by jointly modeling all genes within a locus of interest [5–7]. Similar to marginal TWAS, these methods, with rare exceptions [6], often follow a two-step process. First, they build GReX prediction models for all genes within a genomic locus using the gene expression data [2,8]. Second, after imputing expression into a GWAS dataset, they apply a joint statistical

model to identify the most likely causal gene(s) at that locus. For example, FOCUS (fine-mapping of causal gene sets) extends the Bayesian sparse regression model from GWAS fine-mapping toward TWAS applications [5]. GIFT (gene-based integrative fine-mapping through conditional TWAS) performs conditional TWAS analysis by explicitly adjusting for the GReX of other local genes to fine-map putatively causal genes [6]. cTWAS (causal-TWAS) assesses the focal genes' association with a trait through a Bayesian regression model, while controlling for the effects of nearby genes and variants [7]. By analyzing multiple genes simultaneously, these fine-mapping methods can better dissect complex association signals and distinguish likely causal genes from those that due to mere correlation.

Despite their advantages, current TWAS fine-mapping methods have two important limitations. First, most existing methods rely on gene expression data collected from a single tissue. However, complex diseases often arise from genetic effects that involve multiple tissues, where disease-associated variants influence disease risk by regulating gene expression in a tissue-specific manner [9,10]. Since disease-relevant tissues are often unknown and may not be collected in some studies, focusing solely on one tissue risks overlooking the true causal gene if disease-related activity occurs in unanalyzed tissue(s). In fact, recent studies have shown that integrating GWAS with expression data from multiple tissues can substantially improve the power of the analysis [11]. Second, and perhaps more importantly, most TWAS fine-mapping methods are based on linear regression frameworks, which are specifically designed for quantitative traits like height or blood pressure. When applied to case-control studies, these methods use the same framework, treating binary disease status as continuous values [12,13]. While linear regression models can approximate generalized linear models, such as logistic or probit regression, that are tailored for binary traits, this approach is only accurate when effect sizes are small, as is typical in GWAS [14]. In contrast, GReX in TWAS fine-mapping involves multiple SNP effects, resulting in much larger effect sizes. As a result, directly applying linear models to binary traits in TWAS fine-mapping may fail to accurately capture the binary nature of the outcome, leading to a loss of statistical power and reduced performance for disease traits -- similar to the issues observed in marginal TWAS analysis [15].

Here, we introduce mFABIO (**m**ulti-tissue **F**ine-m**A**pping of causal genes for **BI**nary **O**utcomes),

a novel TWAS fine-mapping method designed to address the limitations discussed above. Specifically, mFABIO utilizes a probit model to directly analyze binary outcomes, jointly models multiple genes within a given locus across multiple tissues, and accounts for potential horizontal pleiotropic effects. We demonstrate the power of mFABIO through comprehensive simulations and by analyzing six binary disease traits from the UK Biobank, using expression data spanning 38 Genotype-Tissue Expression (GTEx) tissues.

## Materials and methods

### Model for multi-tissue TWAS fine-mapping of binary traits

We consider a genome-wide association study (GWAS) of a binary trait conducted on $n$ individuals, where $\boldsymbol{y} \in \{0, 1\}^n$ represents the $n$-vector of binary phenotypes. Our objective is to use transcriptome-wide association study (TWAS) fine-mapping to identify genes whose genetically regulated expression (GReX) is associated with the trait and to prioritize the tissues most relevant to these associations. Different from existing single-tissue TWAS fine-mapping methods [5–7,15], we leverage tissue-specific GReX constructed from multiple tissues and jointly model all gene-tissue pairs within a defined genomic region, while explicitly accounting for the binary nature of the outcome trait. To do so, we consider $K$ different tissues and examine genomic regions one at a time. For each genomic region of focus, we assume there are $M$ genes in total and denote $p_k$ as the number of genes with GReX constructed in tissue $k$ ($p_k \leq M$). These GReX values are assumed to have been pre-computed using standard transcriptome prediction frameworks such as PrediXcan [2]. Let $\hat{\boldsymbol{G}}_{n \times p_k}$ denote the corresponding GReX matrix for these genes across $n$ individuals in tissue $k$. We then define the overall GReX matrix $\hat{\boldsymbol{G}}$ as an $n \times p$ matrix for all $p$ gene-tissue pairs in that region as the concatenation of the $K$ tissue-specific GReX matrices: $\hat{\boldsymbol{G}} := \left[ \hat{\boldsymbol{G}}_{n \times p_1}, \hat{\boldsymbol{G}}_{n \times p_2}, \cdots, \hat{\boldsymbol{G}}_{n \times p_K} \right]_{n \times p}$, where $p = \sum_{k=1}^{K} p_k$. We denote $\boldsymbol{X}$ as an $n \times s$ genotype matrix for the same $n$ individuals, capturing all the $s$ SNPs within the region of focus. We further denote $\boldsymbol{C}$ as an $n \times c$ covariate matrix

for the same $n$ individuals, covering covariates like age, sex, and principal components. To facilitate computation, we center and standardize each column of $\hat{\boldsymbol{G}}$, $\boldsymbol{X}$, and $\boldsymbol{C}$ to have a mean of zero and a standard deviation of one following [16].

To accommodate the binary nature of the trait, we consider the following probit model linking the GReX to the binary outcome:

$$P\left(y_i = 1 \middle| \hat{\boldsymbol{G}}_i, \boldsymbol{X}_i, \boldsymbol{C}_i, \beta, \alpha, \xi\right) = 1 - P\left(y_i = 0 \middle| \hat{\boldsymbol{G}}_i, \boldsymbol{X}_i, \boldsymbol{C}_i, \beta, \alpha, \xi\right) = \Phi\left(\mu + \hat{\boldsymbol{G}}_i^T\beta + \boldsymbol{X}_i^T\alpha + \boldsymbol{C}_i^T\xi\right) \; (i = 1, \cdots, n)$$

Here $y_i \in \{0, 1\}$ is the binary trait for the $i$'th individual; $\hat{\boldsymbol{G}}_i$ is a $p$-vector of GReX for $p$ gene-tissue pairs for the $i$'th individual (i.e., the $i$'th row of $\hat{\boldsymbol{G}}$); $\beta$ is a $p$-vector of corresponding effect sizes for the gene-tissue pairs; $\boldsymbol{X}_i$ is an $s$-vector of genotypes for the $i$'th individual (i.e., the $i$'th row of $\boldsymbol{X}$); $\alpha$ is an $s$-vector of corresponding effect sizes for SNPs, also known as the horizontal pleiotropic effects, capturing the SNPs' effects on the trait not mediated through gene expression; $\boldsymbol{C}_i$ is a $c$-vector of covariates for the $i$'th individual (i.e., the $i$'th row of $\boldsymbol{C}$); $\xi$ is a $c$-vector of corresponding effect sizes for covariates; $\mu$ is a scalar representing the intercept; and $\Phi$ is the cumulative distribution function (CDF) of the standard normal distribution.

The model above can be equivalently re-parameterized using a liability threshold framework. Specifically, following [17], we introduce a latent variable $z_i$ for each individual $i \in \{1, \ldots, n\}$, representing their underlying liability. This yields the following equivalent latent variable representation of Equation (1):

$$y_i = \begin{cases} 1 & \text{if } z_i > 0 \\ 0 & \text{if } z_i \leq 0 \end{cases},$$

(2)

$$z_i = \mu + \hat{\boldsymbol{G}}_i^T\beta + \boldsymbol{X}_i^T\alpha + \boldsymbol{C}_i^T\xi + \varepsilon_i \quad \varepsilon_i \sim N\left(0, 1\right).$$

(3)

The above formulation explicitly distinguishes between the effects mediated by tissue-specific gene expression ($\hat{\boldsymbol{G}}_i^T\beta$) and those arising from horizontal pleiotropy ($\boldsymbol{X}_i^T\alpha$) and other covariates ($\boldsymbol{C}_i^T\xi$). Among them, the horizontal pleiotropic effects from SNPs are captured by the effect size vector $\alpha$, which represents the aggregate direct effects of genetic variants on the phenotype that are not mediated by the predicted expression features in $\hat{\boldsymbol{G}}$. While explicit inclusion of the genotype matrix $\boldsymbol{X}$ accounts for the correlation structure among SNPs, we further model the potential correlation among elements of $\alpha$ using a Bayesian ridge regression framework. Specifically, we place an independent and identically distributed (i.i.d.) Gaussian prior on the SNP-effect vector $\alpha$ across the $s$ SNPs following [18]:

$$\alpha \sim N\left(\boldsymbol{0}, \sigma_\alpha^2 \boldsymbol{I}_s\right),$$

(4)

where we further assume that $\sigma_\alpha^2$ follows a conjugate and weakly informative inverse-gamma prior $\sigma_\alpha^2 \sim IG(1, 1)$. By using a ridge penalty induced by the Gaussian prior, the model robustly handles the high multicollinearity inherent in genotype data, while ensuring that our variable selection mechanism (SuSiE) remains focused on the primary features of our interest, the gene-tissue pairs.

For the intercept $\mu$, we assume a normal distribution: $\mu \sim N(0, \sigma_\mu^2)$, where $\sigma_\mu^2 \to \infty$ as a non-informative prior. For the effect sizes $\xi$ on covariates, we also assume a normal distribution: $\xi \sim N\left(\boldsymbol{0}, \sigma_\xi^2 \boldsymbol{I}_c\right)$, where $\sigma_\xi^2 \to \infty$. Both of them follow the common practice in Bayesian approaches, which is equivalent to treating them as a fixed effect.

### Decomposition of gene-tissue pair effects

Our goal is to identify the potential causal gene-tissue pair underlying the trait through inferring the tissue-specific posterior probability that the GReX of each gene is associated with the trait. A critical challenge in identifying causal

signals is the high correlation among the predicted expression levels (GReX) of genes located in the same linkage disequilibrium (LD) region. Standard marginal association approaches often implicate large blocks of correlated genes, obscuring the true causal drivers. Our model resolves this ambiguity by assuming only a small subset of the $p$ possible gene-tissue pairs displays non-zero effects. To capture this sparse structure, we adopt the Sum of Single Effects (SuSiE) prior [19] for the gene-tissue effect sizes $\beta$ in Equation (3), which is decomposed as a sum of $L$ single-effect components:

$$\beta := \sum_{l=1}^{L} \beta_l \gamma_l.$$

(5)

Here we assume that there are at most $L$ gene-tissue pairs in the region with non-zero effects ($L$ is set to be 10 in the present study following [19]). Each $\beta_l \gamma_l$ term is a single-effect component, where $\beta_l$ is a scalar effect-size parameter following a normal prior $N\left(0, \sigma_\beta^2\right)$, for $l = 1, 2, \ldots, L$. For the $l$'th single-effect component ($l \in \{1, 2, \ldots, L\}$), we use a $p$-vector of binary indicators $\gamma_l$ to indicate which gene-tissue pair is responsible for the $l$'th component: the element in $\gamma_l$ that corresponds to the $l$'th non-zero effects is 1 while all the other elements in $\gamma_l$ are 0's. We further assume a prior distribution $\gamma_l \sim Multinomial\left(1, \pi\right)$, allowing each single effect to correspond to a specific column of $\hat{G}$. Here, the prior non-zero probability for each element of $\gamma_l$ is determined by each element of the $p$-vector $\pi := (\pi_1, \ldots, \pi_p)$. The structure of $\pi$ will be introduced in the following section.

By modeling the total GReX as a sum of $L$ distinct, sparse components, the framework forces correlated gene-tissue pairs to compete for inclusion in the model. Importantly, inference is performed directly on the design matrix using regularized updates, ensuring numerical stability even when predictors are highly collinear or rank-deficient. During the variational inference procedure (will be introduced in the following sections), the posterior probability assigned to any specific gene-tissue pair is calculated conditional on the effects of all other selected features. This conditional updating, together with the sparsity-inducing SuSiE prior, enables the model to effectively fine-map the signal by assigning high posterior probability only to those pairs that provide additional explanatory power beyond correlated alternatives. In cases of extreme collinearity (e.g., nearly identical predictors across tissues), the model appropriately distributes posterior inclusion probability across indistinguishable features rather than producing unstable estimates, while shrinking the effects of non-causal neighbors that are correlated due to LD toward zero.

## Hierarchical prior for variable selection

To infer causal probabilities at both the gene and gene-tissue pair levels, we first define $\pi_j$ as the $j$'th element of the $p$-vector $\pi$ as defined in the preceding section. Here, the $j$'th index corresponds to the specific pair of the $k$'th tissue and the $m$'th gene, thus $j \in \{1, \cdots, KM\}$ for $K$ tissues and $M$ genes. The prior probability $\pi_j := P(\gamma_{lj} = 1)$, represents the probability that the $j$'th gene-tissue pair is responsible for the $l$'th single-effect component. We define this prior in a hierarchical fashion. Specifically, following [20], we first denote the vector of prior probabilities for selecting each of the $M$ unique genes as an $M$-vector $\pi_G = (\pi_{g_1}, \ldots, \pi_{g_M})$. For the $m$'th unique gene $g_m$, we then denote the corresponding vector of prior probabilities for selecting each tissue given that the gene $g_m$ has been selected as a $K$-vector $\pi_{T|g_m} = (\pi_{t_1|g_m}, \ldots, \pi_{t_K|g_m})$. Afterwards, we assume a *priori* on $\pi_j$ as:

$$\pi_j = P\left(select\ pair\ \{g_m, t_k\}\right)$$

$$= P\left(select\ gene\ g_m\right) \times P\left(select\ tissue\ t_k \middle|\ gene\ g_m\ is\ selected\right)$$

$$= \pi_{g_m} \times \pi_{t_k|g_m}$$

(6)

where $\sum_{j=1}^{p} \pi_j$ is ensured to be 1 under this structure.

We further specify Dirichlet priors on the gene-level probability vector $\pi_G$ and on each of the tissue probability vectors $\pi_{T|g_m}$: $\pi_G \sim Dirichlet(1,\ldots,1)$ and $\pi_{T|g_m} \sim Dirichlet(1,\ldots,1)$, to represent an uninformative starting belief that all genes, and all tissues within a gene, are equally likely to be selected. This choice is statistically motivated for two primary reasons. First, the Dirichlet distribution's support is the probability simplex, which is the valid domain for probability vectors that must sum to one. Second, and critically for computational feasibility, it is the conjugate prior for the categorical selection process. This property of conjugacy ensures that the posterior distributions of the probability vectors remain in the Dirichlet family, making their estimation tractable within our following variational Bayes inference, and allowing the model to learn data-driven, gene- and tissue-specific probabilities empirically.

## Scalable variational inference

With the above model settings, our goal is to infer the posterior probability that each element of $\gamma_l$ equals one. This is achieved through the posterior estimates of the nested vectors $\pi_G$ and $\pi_{T|*}$, which together enable the identification of causal genes, and subsequently, the causal tissues underlying these genes. To ensure scalable inference, we developed a variational Bayes algorithm. Specifically, we denote $\theta$ as the set of hyperparameters defined in the preceding section, where $\theta = (\sigma_\mu^2, \sigma_\alpha^2, \sigma_\beta^2, \sigma_\xi^2)$, and $z$ as the latent vector containing all $z_i$'s defined in Equations (2) and (3). We then employ an empirical Bayes approach to obtain an estimate of $\hat{\theta}$ (details in the S1 Text). With the estimated $\hat{\theta}$, we approximate the complex conditional posterior distribution $p\left(z,\mu,\beta,\alpha,\xi \mid \hat{G},X,C,y,\hat{\theta}\right)$ using an approximate conditional posterior distribution of $q\left(z,\mu,\beta,\alpha,\xi \mid \hat{G},X,C,y,\hat{\theta}\right)$.

We first assume that the approximate conditional posterior distribution $q\left(z,\mu,\beta,\alpha,\xi\right)$ is in the form of the following factorization:

$$q\left(z,\mu,\beta,\alpha,\xi\right) = q\left(z\right) \times q(\mu) \times q\left(\alpha\right) \times q\left(\xi\right) \times \prod_{l=1}^{L} q\left(\beta_l\right) q\left(\gamma_l\right)$$

$$= \left(\prod_{i=1}^{n} q(z_i)\right) \times q(\mu) \times q(\alpha) \times q(\xi) \times \left(\prod_{l=1}^{L} q(\beta_l) q(\gamma_l|\pi_G, \pi_{T|*}) q(\pi_G) \left(\prod_{*=g_1}^{g_M} q(\pi_{T|*})\right)\right) \tag{7}$$

This equation specifies the structural decomposition of the approximate conditional posterior distribution $q\left(z,\mu,\beta,\alpha,\xi\right)$, encapsulating the mean-field independence assumptions required to render the inference computationally tractable. Following the common practice in Bayesian approaches for the simplicity of the notations, we omitted the conditioning on $(\hat{G},X,C,y,\hat{\theta})$ in Equation (7) as well as in the following text. Observed data like $\hat{G}$, $X$, and $C$ enter the model through the coordinate ascent variational inference (CAVI) updates, where the optimal parameter values for each factor are derived by maximizing the evidence lower bound (ELBO).

Here, Equation (7) contains eight components: $q(z_i)$, $q(\mu)$, $q(\alpha)$, $q(\xi)$, $q(\beta_l)$, $q(\gamma_l|\pi_G, \pi_{T|*})$, $q(\pi_G)$, and $q(\pi_{T|*})$. For the approximate posterior $q(z_i)$, we assume it follows a truncated normal distribution, whose parameters are updated based on the observed binary trait $y_i$. For the approximate posterior $q(\mu)$, we assume it follows a normal distribution, given $\mu$ has a normal prior as well as the normal-normal conjugation. Similarly, for the approximate posterior $q(\alpha)$ and $q(\xi)$, we assume each of them follows a multivariate normal distribution, given its multivariate normal prior defined earlier. For the approximate posterior $q(\beta_l)$, we assume it also follows a normal distribution for $l = 1, 2, \ldots, L$. For the approximate posterior $q(\gamma_l|\pi_G, \pi_{T|*})$, we assume it is in the form of a multinomial distribution, with

$$q\left(\gamma_l|\pi_G, \pi_{T|*}\right) \sim Multinomial(1, \widetilde{\pi}_l), \tag{8}$$

where the $p$-vector $\widetilde{\pi}_l = (\widetilde{\pi}_{l1}, \cdots, \widetilde{\pi}_{lp})$ represents the posterior probability that each gene-tissue pair represents the $l$'th non-zero effect, and the corresponding estimate is known as the posterior inclusion probability (PIP). For the approximate posteriors $q(\pi_G)$ and $q(\pi_{T|*})$, both are assumed to follow the Dirichlet distributions, and the posterior Dirichlet distribution $q(\pi_G)$ over the $M$ unique genes is used to obtain the gene-level PIP. With the specified form of the approximate posterior $q(z, \mu, \beta, \alpha, \xi)$, we aim to minimize its Kullback-Leibler (KL) divergence with the actual posterior $p(z, \mu, \beta, \alpha, \xi)$. Minimizing the KL divergence is equivalent to maximizing the evidence lower bound (ELBO), which can be achieved by an iterative Bayesian stepwise selection algorithm following the strategy of SuSiE [19]. The detailed inference steps for each parameter are provided in the S1 Text.

**Calculation of posterior inclusion probability (PIP)**

To obtain the association evidence of each gene-tissue pair, we calculate the posterior probability that the $j$'th gene-tissue pair has a non-zero effect, or $PIP_j$, which equals to that the $j$'th gene-tissue pair is selected by at least one of the $L$ single-effect components. Within our variational approximation, we assume the selections for each component are independent in the posterior. Thus, we first calculated the posterior probability that none of the $L$ single-effect component selects the $j$'th gene-tissue pair:

$$q(\text{the } j\text{'th pair is not selected}) = \prod_{l=1}^{L} \left(1 - \widetilde{\pi}_{lj}\right),$$

(9)

and the PIP for the $j$'th gene-tissue pair can be calculated as:

$$PIP_j = 1 - \prod_{l=1}^{L} \left(1 - \widetilde{\pi}_{lj}\right), \quad j = 1, \ldots, p$$

(10)

To obtain the association evidence of each gene, we calculate the overall posterior probability that the gene $g_m$ is involved in the phenotype through at least one of its tissues, or $PIP_{g_m}$. Within our variational approximation, the parameter vector $\pi_G$ represents the probabilities of selecting each gene, so we can naturally estimate the gene-level PIP using the posterior mean of the corresponding gene selection probability. Specifically, assuming the final approximate posterior distribution for $\pi_G$ is: $q(\pi_G) = Dirichlet(\widetilde{\delta}_{g_1}, \ldots, \widetilde{\delta}_{g_M})$, and $\pi_G := (\pi_{g_1}, \ldots, \pi_{g_M})$, we have:

$$PIP_{g_m} = E_q\left[\pi_{g_m}\right] = \frac{\widetilde{\delta}_{g_m}}{\sum_{m'=1}^{M} \widetilde{\delta}_{g_{m'}}}, \quad m = 1, \ldots, M$$

(11)

The PIP in our model serves as an important measure of evidence for the gene or gene-tissue pair associated with the outcome trait. We refer to our method and inference algorithm as the **m**ulti-tissue **F**ine-m**A**pping of causal genes for **BI**nary **O**utcomes (mFABIO). mFABIO is implemented as an R package, freely available at https://superggbond.github.io/mFABIO/.

**Simulations**

We conducted simulations to assess the performance of mFABIO and compare it with four single-tissue TWAS fine-mapping methods (FABIO, FOCUS, GIFT, cTWAS) and one multi-tissue method (TGFM). We used real genotype data from the UK Biobank, simulating both gene expression and a binary outcome trait. Specifically, we randomly sampled 50,000–250,000 individuals of European ancestry in UK Biobank to serve as the GWAS data and another 500–10,000

individuals of European ancestry (disjoint from GWAS samples) to serve as the eQTL mapping data. Our simulations focused on 426,593 SNPs on chromosome 1, which were partitioned into 133 independent LD blocks using LDetect [21], following the same procedure of [5–7,15]. From these, we randomly selected 50 LD blocks, each containing at least five genes. Within these selected blocks (containing 5–43 genes per block; mean = 16, median = 13), we extracted cis-SNPs located within 100 kb upstream of the transcription start site (TSS) and 100 kb downstream of the transcription end site (TES) of each gene, following [6,15]. We retained 789 genes that had at least 10 cis-SNPs, and these were used for the following simulations.

We then simulated the genetic architecture of gene expression across multiple tissues in both the GWAS and eQTL mapping panels. Following [11], we simulated all 789 selected genes to be expressed in three tissues and designated 50% of these gene-tissue pairs as cis-genetically heritable. For each heritable pair, we randomly assigned five causal cis-SNPs: three shared across tissues and two specific to each tissue, and assumed each causal cis-SNP to explain a fixed proportion of gene expression variance. Following [15,22], we denoted the proportion of gene expression variance explained by genetic effects for each heritable gene-tissue pair as $PVE_1$. While all expressed gene-tissue pairs were considered during fine-mapping, only these heritable pairs were assigned simulated causal effects on the final trait. In the eQTL mapping panel, we applied SuSiE to the corresponding real genotypes and simulated gene expression data to generate prediction models for each gene-tissue pair.

We simulated the binary trait in the GWAS panel using a logistic regression model based on the simulated gene expression data. The probability of the binary trait ($y_i$ = 1) for the $i$'th individual was determined by a latent variable $\eta_i$ as:

$$P\left(y_i = 1 \middle| \eta_i\right) = \frac{1}{1 + \exp\left(-\eta_i\right)}. \tag{12}$$

The latent variable $\eta_i$ links the trait to the predicted gene expression and is defined as a weighted sum of causal gene effects across three tissues, following the structure from [23]:

$$\eta_i = u + w_1\hat{g}_{1i}^T v_1 + w_2\hat{g}_{2i}^T v_2 + w_3\hat{g}_{3i}^T v_3 + \epsilon_i, \quad \epsilon_i \sim N\left(0, 1 - \left[n_g \cdot PVE_2\right]\right), \tag{13}$$

where $u$ is an intercept that controls the case:control ratio as it equals to the expectation of log(case#/control#); $w_1$, $w_2$, $w_3$ are the contributing weights for each tissue, with $w_1 + w_2 + w_3 = 1$; $\hat{g}_{*i}$ is an $n_g$-vector of simulated gene expression of the selected causal gene-tissue pairs for the $i$'th individual in the corresponding tissue. For each tissue, we randomly selected $n_g$ causal gene-tissue pairs from the heritable pairs, and denote the proportion of the phenotype's variance explained by each causal gene-tissue pair as $PVE_2$, following [15,22]. And $v_*$ is an $n_g$-vector of genes' causal effect sizes in the corresponding tissue, with each element $v_{*j}$ set to be: $v_{*j} = \sqrt{\frac{PVE_2}{PVE_1}}$, following [15,22]. Notably, we intentionally used a logistic regression model instead of a probit model to assess our method's performance under model misspecifications.

In the simulations, we first examined the calibration of test statistics from different methods under the null simulation settings, where we set $PVE_2$ = 0% so that none of the gene-tissue pairs were associated with the trait. Besides the null simulations, we also examined the power of different methods on detecting the causal genes among tissues under the alternative simulation settings with non-zero $PVE_2$. For both the null and alternative simulations, we started from a baseline simulation setting and then varied one parameter at a time on top of the baseline setting to examine the influence of different parameters. Specifically, for $PVE_1$, we set it to be 5%, 7.5%, 10% (baseline setting), following the same settings of [11]. For the number of causal cis-SNPs for each heritable gene-tissue pair, we set it to be either 5 (baseline setting), or a random number between 3 and 7 (inclusive) following [11]. For $PVE_2$, we varied its value to be either 0% (null setting), 0.2%, 0.4% (baseline setting), or 0.6% following [15,22]. For the number of causal genes per tissue ($n_g$), we fixed it to be 10 following [15]. In terms of the contributing weights for each tissue, we considered three settings following [23]:

(1) $w_1 = 1$, $w_2 = w_3 = 0$; (2) $w_1 = w_2 = \frac{1}{2}$, $w_3 = 0$ (baseline setting); and (3) $w_1 = w_2 = w_3 = \frac{1}{3}$. For the binary trait, we examined different case:control ratios by varying the intercept $u$, using 1:1 as the baseline setting ($u = 0$) while exploring other ratios of 1:10 ($u = \log(0.1)$), and 1:50 ($u = \log(0.02)$). Furthermore, we extended the baseline setting to incorporate horizontal pleiotropy, allowing SNPs to affect the trait independently of the modeled gene expression. Following [24], we assumed a total phenotypic heritability of 0.1, and set the variance explained by GReX effects to 4% same as our baseline setting, distributed as 0.4% per pair across 10 causal gene-tissue pairs. Consequently, the remaining genetic variance (6%) was assigned to SNP horizontal pleiotropy. To model these effects, we included all SNPs within the LD region, consistent with the ridge regression assumptions outlined in Equations (2)-(4). We also evaluated how the sample sizes of the GWAS study panel and the eQTL reference panel affect the performance of the methods. Specifically, we varied the GWAS sample size from 50K (baseline) to 100K, 150K, 200K, and 250K, and the eQTL reference sample size from 500 (baseline) to 1K, 5K, and 10K, while keeping all other parameters fixed at their baseline values.

We performed 100 simulation replicates for each simulation setting. In each replicate, we applied all methods, except TGFM, to analyze one LD block at a time following their design. TGFM was applied to each chromosome, as its software is specifically designed to internally partition chromosome-level data into overlapping 3 Mb loci for analysis, and modifying it to analyze LD blocks defined differently is technically challenging. For multi-tissue methods (mFABIO and TGFM), we directly applied them on gene-tissue pairs, while for single-tissue methods (FABIO, FOCUS, GIFT, and cTWAS), we applied them in two ways: (1) applied them on each tissue with corresponding tissue-specific predicted gene expression levels following their original designs, then combined the results across tissues; (2) naturally extended their designs to analyze gene-tissue pairs jointly, by preparing the gene-tissue pair level input files of cis-predicted expression models, and then applied them on gene-tissue pairs. We refer to the resulting methods in (2) as FABIO-gt, FOCUS-gt, GIFT-gt, and cTWAS-gt, respectively. In either approach, we summarized the results across simulation replicates. In the analysis, we first examined the calibration of test statistics from different methods under the null simulation settings, by examining the number of false discoveries at an estimated false discovery rate (FDR) of 0.05. The calculation of the estimated FDR will be introduced in the following paragraph. We then evaluated the power of different methods at the FDR of 0.05 under the alternative simulation settings. Since we knew the ground truth in the simulations, we first compared the power of different methods under a true FDR threshold of 0.05. Specifically, to compute the true FDR in the simulations, we first divided 100 simulation replicates into five groups, with 20 replicates per group. In each group, we obtained the test statistics from each method for all gene-tissue pairs across the 20 replicates, either in the form of p-values or PIPs. For each method in turn, we sorted the test statistics from the most significant (i.e., the smallest p-value or the largest PIP) to the least significant (i.e., the largest p-value or the smallest PIP). For each ordered test statistic in the sorted list, we counted the number of false positives detected above the threshold defined by this specific test statistic. The number of false positives, divided by the total number of gene-tissue pairs called significant at that threshold, is the true FDR. We then determined threshold that corresponds to a true FDR of 0.05, based on which we further calculated power. Power is calculated as the number of true positives above the threshold in the sorted list divided by the total number of causal gene-tissue pairs. Afterwards, we obtained the average power across five groups to minimize stochasticity [6,15,22].

We also compared the methods at gene level. For multi-tissue methods mFABIO and TGFM that output gene-level PIPs, we directly used the same way to calculate the power at a true FDR of 0.05, based on the gene-level PIPs. For single-tissue methods FABIO, FOCUS, and cTWAS that only output PIPs at gene-tissue pair level, we first estimated the gene-level PIP by calculating the probability that the gene is causal in at least one tissue with the assumption of independence: $PIP_{gene} = 1 - \prod_m (1 - PIP_m)$. Then we calculate the power at a true FDR of 0.05, based on the estimated gene-level PIPs. For single-tissue method GIFT that only outputs p-values at gene-tissue pair level, we first estimated the gene-level p-values using Fisher's method [25] to combine the p-values with the assumption of independence. Then we calculated the power at a true FDR of 0.05, based on the estimated gene-level p-values.

Because we can only compute the true FDR in the simulations but not in the real data, we also compared the power of different methods under an estimated FDR threshold of 0.05. We estimated FDR using different approaches: the Benjamini-Hochberg approach [26] to estimate FDR using p-values from the frequentist method (GIFT), and the local FDR approach [27] to estimate FDR using PIPs from the Bayesian methods (mFABIO, TGFM, FABIO, FOCUS, and cTWAS). Specifically for the local FDR approach, we first calculated the local FDR for all the gene-tissue pairs, which is effectively 1-PIP [6,15]. We then sorted gene-tissue pairs based on these local FDRs from small to large. For each gene-tissue pair in turn, we estimated FDR for that pair by aggregating the local FDRs from the first pair up to that pair in the sorted list. Afterwards, we identified the cutoff value of PIP/local FDR that corresponds to a targeted FDR threshold of 0.05. This is the significant threshold we use that corresponds to an estimated FDR of 0.05. We also compared the methods at gene level, by first estimating the gene-level PIPs or p-values as illustrated previously, and then we calculated the power at an estimated FDR of 0.05.

## Real datasets

**Genotype-Tissue Expression (GTEx) data.** We constructed the genetically regulated expression (GReX) using publicly available cis-predicted expression models (https://doi.org/10.7910/DVN/8IPOPK) derived from the Genotype-Tissue Expression (GTEx) project [11]. These models were built using data from individuals of European ancestry. To mitigate heterogeneity in expression quantitative trait loci (eQTL) sample sizes, the original 47 GTEx tissues were aggregated into 38 meta-tissues with more homogeneous sample sizes. For each meta-tissue, models included only protein-coding genes, with cis-eQTLs defined as variants within 500 Kb of a gene's transcription start site (TSS). A series of quality control filters were applied by removing variants with a minor allele frequency (MAF) < 0.05, strand-ambiguous variants, and those not present in the UK Biobank dataset. This filtering resulted in a final set of 10,545,304 variants for model building. Prediction models for each gene-tissue pair were then generated by applying SuSiE [19] to the eQTL summary statistics and in-sample linkage disequilibrium (LD) matrices. More details on the model construction can be found in [11]. Ultimately, this process yielded models for 799–5,496 genes per meta-tissue, comprising a total of 13,700 unique genes and 119,270 unique gene-tissue pairs.

**UK Biobank (UKBB) data.** The UK Biobank data consists of 487,409 individuals and 92,693,895 imputed autosome SNP variants [28]. We followed the same sample QC procedure as Neale Lab (https://github.com/Nealelab/UK_Biobank_GWAS/tree/master/imputed-v2-gwas) to retain a total of 337,198 individuals of European ancestry. To ensure compatibility with the cis-predicted expression models described previously, the UKBB genotype data was then filtered to retain only the 10,545,304 variants used to build those models.

## Real data application

We performed fine-mapping on six binary disease traits through integrating the GTEx cis-predicted expression models on different meta-tissues with the UKBB genotype data. The examined disease traits include asthma, breast cancer, gout, hypertension, prostate cancer, and rheumatoid arthritis. These disease traits were selected following [15,29] and have a wide range of prevalence (0.02 to 0.38). For the two sex-related traits (breast cancer and prostate cancer), we limited our analysis to female individuals (for breast cancer) or male individuals (for prostate cancer). For a given trait in UKBB, we kept all individuals of European ancestry with self-reported case status of that trait in the GWAS data. Same as the simulations, SuSiE-based cis-predicted expression models were used for all the comparing methods (mFABIO, TGFM, FABIO, FOCUS, GIFT, and cTWAS). For the multi-tissue methods mFABIO and TGFM, we directly used the cis-predicted expression models from all 38 meta-tissues. In contrast, for the single-tissue methods (FABIO, FOCUS, GIFT, and cTWAS), we naturally extended their designs to analyze gene-tissue pairs jointly, a strategy that showed superior performance in simulations over single-tissue analyses. Same as the simulations, we prepared the gene-tissue pair level input files of cis-predicted expression models, and then applied them on gene-tissue pairs. The genotype input data was then prepared

based on each method's design. For mFABIO and FABIO, which are built for binary outcomes, we used individual-level GWAS genotypes. For the other four methods (TGFM, FOCUS, GIFT, and cTWAS), which require summary statistics, we generated these by performing a logistic regression on the GWAS data using PLINK 2.0 [30], adjusting for the top 10 genotype PCs and sex. We then paired either the individual-level GWAS or the GWAS summary statistics with the variant weights from cis-predicted expression models as input, and analyzed 1,433 independent LD blocks [21] across all 22 autosomal chromosomes one at a time, except for TGFM. We applied TGFM to 2,682 overlapping 3 Mb loci spanning the entire genome following its own design. We also performed marginal TWAS analysis for each tissue-disease pair using FUSION [1] with the same GWAS summary statistics and the variant weights. For a fair comparison across methods, we used an estimated FDR threshold of 0.05 for all methods to declare significance, as described in detail in the simulations.

To assess the robustness of mFABIO when eQTL reference panels originated from independent cohorts, we used the same UK Biobank hypertension dataset and replaced the GTEx whole blood eQTL panel (n = 320) with a pseudobulk PBMC eQTL panel (n = 113; SNP weights available at https://doi.org/10.7910/DVN/8UL8XB), while keeping all other tissue models from GTEx unchanged. To ensure comparability, the PBMC panel was restricted to European-ancestry samples and harmonized to the same genomic reference build and variant set used in GTEx. Gene expression normalization procedures were matched to those used in GTEx, and allele alignment and variant harmonization were performed by the panel's publisher to ensure consistency with the GTEx reference models. This resulted in 628 genes modeled in both panels, comprising 538,801 SNPs.

To validate our findings, we further evaluated whether significant gene-tissue pairs (estimated FDR < 0.05) were enriched in tissues with known disease relevance. We established this relevance by querying PubMed for publications over the last 10 years for each disease-tissue combination. The three GTEx tissue categories with the highest publication counts for each disease were designated potentially "disease-relevant". This literature review yielded the following classifications:

- asthma: lung, skin (GTEx tissues: Skin_Sun_Exposed_Lower_leg and Skin_Not_Sun_Exposed_Suprapubic), and blood (GTEx tissue: Whole_Blood)

- breast cancer: breast mammary tissue, blood (GTEx tissue: Whole_Blood), and liver

- gout: blood (GTEx tissue: Whole_Blood), liver, and skin (GTEx tissues: Skin_Sun_Exposed_Lower_leg and Skin_Not_Sun_Exposed_Suprapubic)

- hypertension: blood (GTEx tissue: Whole_Blood), heart (GTEx tissues: Heart_Atrial_Appendage and Heart_Left_Ventricle), and artery (GTEx tissues: Artery_Aorta, Artery_Coronary, and Artery_Tibial)

- prostate cancer: prostate, blood (GTEx tissue: Whole_Blood), and liver

- rheumatoid arthritis: blood (GTEx tissue: Whole_Blood), fibroblast (GTEx tissue: Cells_Cultured_fibroblasts), and spleen

Based on this classification, we generated the absolute counts and calculated the proportion of each method's significant gene-tissue pairs that fell within the relevant tissues for a given disease.

## Results

### Method overview

mFABIO is described in the Materials and Methods section, with technical details provided in the S1 Text and a method schematic shown in Fig 1.

Briefly, mFABIO is a multi-tissue TWAS fine-mapping method designed to identify gene-tissue pairs whose GReX is associated with a binary trait of interest. Different from existing TWAS fine-mapping approaches, mFABIO explicitly models the binary nature of the outcome trait through a probit model and jointly analyzes genes within a given region across

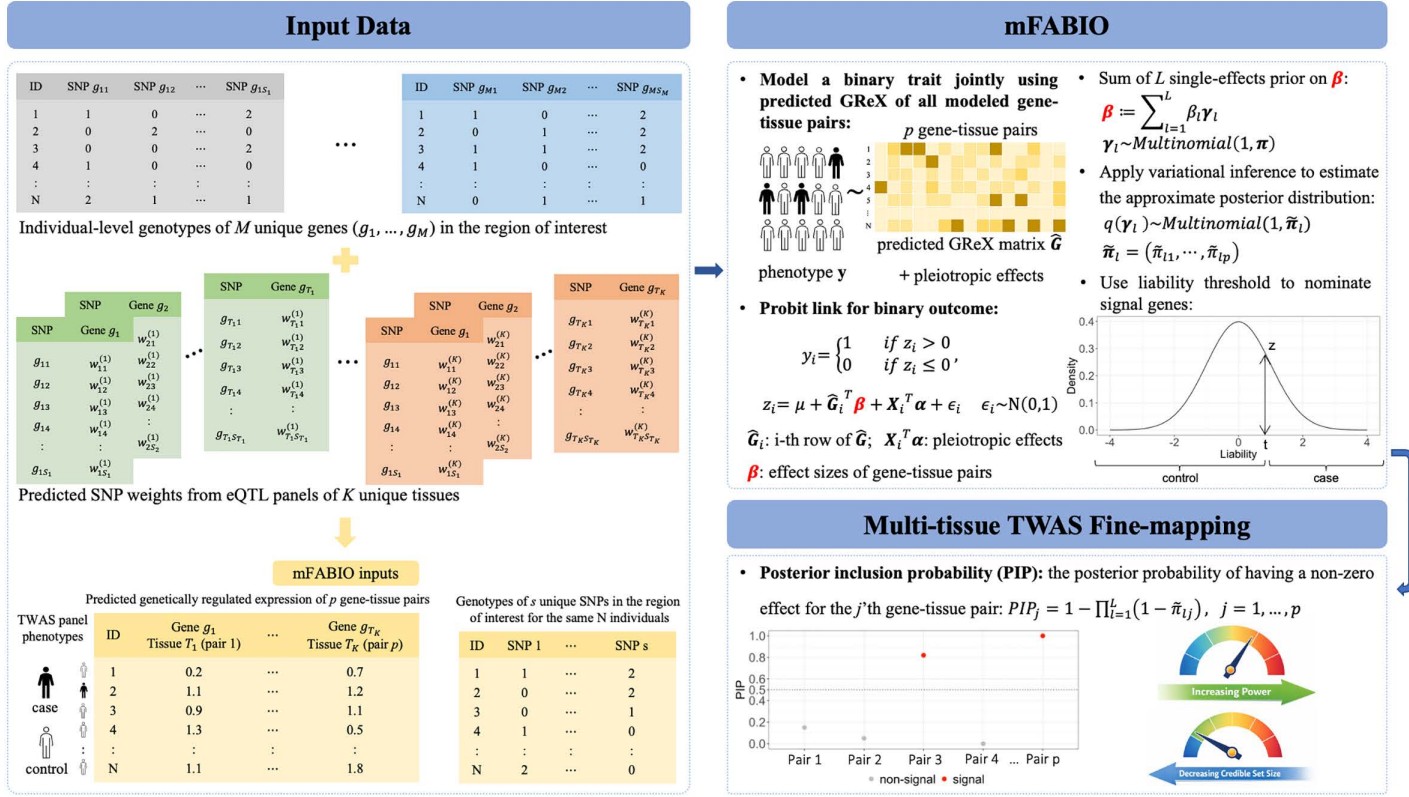

**Fig 1. Schematic overview of mFABIO for multi-tissue TWAS fine-mapping of binary outcomes.** Similar to other existing single-tissue and multi-tissue methods, mFABIO is a two-step TWAS fine-mapping approach that requires the predicted genetically regulated expression (GReX) in the study cohort as input. Shown under the "Input Data" section, mFABIO analyzes one genomic region at a time. For each genomic region containing $s$ SNPs and $M$ genes, mFABIO begins with individual-level genotypes of $s$ SNPs within the region of interest. It also takes as input the SNP prediction weights for each of the $M$ genes, which can be obtained from eQTL mapping cohorts across different tissues using standard methods such as SuSiE. Because each tissue may include a different subset of the $M$ genes, the gene notations $g_1$, $g_2$, . . . , $g_{T_K}$ in this figure are tissue-specific. The term $w_{T_K S_{T_K}}^{(K)}$ denotes the weight of the $S_{T_K}$-th SNP for the $T_K$-th gene in the K-th tissue. These genotypes and SNP weights are then combined to generate predicted GReX for the $p$ gene-tissue pairs in the study cohort. Additionally, mFABIO requires the genotypes of all $s$ SNPs used in the GReX modeling to account for potential horizontal pleiotropic effects. Shown under the "mFABIO" section, mFABIO explicitly models the binary nature of the outcome trait through a latent variable $z$ with a sum of $L$ single-effects prior on the gene-tissue pair effect sizes $\beta$. It also simultaneously models all gene-tissue pairs on a region of interest to account for the GReX correlation, through the input of individual-level GReX matrix. We apply variational inference to estimate the approximate posterior distribution of the single-effect selection vector $\gamma_l$, and use the posterior inclusion probability (PIP) as the evidence for the gene-tissue pair associated with the binary outcome trait ("Multi-tissue TWAS Fine-mapping" section). * Icons in this figure were generated using ChatGPT (OpenAI).

multiple tissues. This joint modeling accounts for GReX correlations among genes and tissues to enhance the power and precision of TWAS fine-mapping. We adapt the variational inference within an iterative Bayesian stepwise selection algorithm of SuSiE, making mFABIO scalable to large biobank scale datasets.

### mFABIO controls false signals in null simulations

We first assessed the false positive rates of all methods across eight null simulation settings ($PVE_2$ = 0%, and other parameters were varied one at a time on top of the baseline setting) using an estimated FDR threshold of 0.05. All methods demonstrated well-controlled false positives, identifying fewer than 0.50 false signals on average per replicate at both

the gene-tissue pair (S1a Fig) and gene levels (S1b Fig). The multi-tissue methods, mFABIO and TGFM, were particularly stringent, showing no false positives under most settings. The only exceptions were an average of 0.20 false-positive gene-tissue pairs and 0.10 false-positive genes when the case:control ratio was 1:50 for both mFABIO and TGFM, and an average of 0.10 false-positive gene-tissue pairs and 0.10 false-positive genes when there were 3 causal tissues for TGFM; all well below the expected one ($0.05 \times 20$ replicates per simulation group) false signal. The single-tissue methods also exhibited low error rates. On average, their false positive identifications ranged from 0.10 to 0.23 at the gene-tissue pair level and from 0.08 to 0.18 at the gene level across all null scenarios.

## mFABIO produces calibrated PIPs and improves power for fine-mapping causal signals in alternative simulations

We then evaluated how calibrated the PIPs/p-values are from different methods in terms of controlling the false discovery rate (FDR) at an estimated FDR threshold of 0.05 in our alternative simulations. Under the baseline setting at the gene-tissue pair level, the actual FDR for multi-tissue methods was well-calibrated to this threshold, with mFABIO and TGFM both at 0.04. In contrast, the actual FDR for all single-tissue methods (FABIO-gt, FABIO, FOCUS-gt, FOCUS, GIFT-gt, GIFT, cTWAS-gt, and cTWAS) was inflated, ranging from 0.13 to 0.16 (S2a Fig). Similar results were observed under different case:control ratios (S2a Fig) and numbers of causal tissue(s) (S2b Fig), as well as other simulation settings (S2c–S2f Fig): across all settings, the actual FDR that corresponds to the estimated FDR of 0.05 was on average 0.04 for both mFABIO and TGFM, while ranged from 0.12 to 0.15 for single-tissue methods. The superiority of the multi-tissue approach was also evident at the gene level, where the average true FDR under an estimated FDR threshold of 0.05 was 0.04 for mFABIO and TGFM, versus 0.11 to 0.14 for single-tissue methods (S3a-S3f Fig).

Next, we evaluated the power of different methods in detecting causal gene-tissue pairs and genes under alternative settings. To ensure a fair comparison that accounts for the previously observed FDR inflation, we first computed the power based on a true FDR threshold of 0.05. We found that mFABIO is more powerful than other methods under all alternative simulation settings. For example, in the baseline setting at a true FDR of 0.05, mFABIO substantially outperformed the other approaches. For identifying causal gene-tissue pairs, mFABIO achieved a power of 34.5%, compared to 27.5% for TGFM and a range of 15.0%-19.0% for the single-tissue methods (Fig 2a). This trend held for identifying causal genes, where mFABIO's power was 36.3%, versus 28.9% for TGFM and 15.8%-19.5% for the single-tissue methods (Fig 2b).

We carefully examined the influence of different simulation parameters on the power of different methods. In terms of case:control ratio, we found that the power decreased for all methods as the case:control ratio became more imbalanced. However, the relative performance ranking remained consistent, with the two multi-tissue methods outperforming the single-tissue methods and mFABIO ranking as the most powerful in all conditions. For instance, at the gene-tissue pair level, as the ratio decreased from 1:1–1:10 and 1:50, mFABIO's power went from 34.5% to 29.0% and 24.5%. Following the same trend, TGFM's power decreased from 27.5% to 23.0% and 19.5%, while the power for single-tissue methods dropped from a range of 15.0%-19.0% down to 11.0%-13.0% (Fig 2a). A similar trend was observed at the gene level (Fig 2b). Next, we investigated how the number of causal tissues affects the power, and we found that the power of all methods decreased as the number of causal tissues increased from one to three. This was an expected result of our simulation design, as the total effect size across causal gene-tissue pairs was fixed, and scenarios with more causal tissues had a smaller effect size for each pair. Despite this overall power reduction, the performance ranking of different methods remained consistent. At the gene-tissue pair level, mFABIO was the most powerful, with its power decreasing from 41.0% (with one causal tissue) to 29.3% (with three). For comparison, TGFM's power dropped from 35.0% to 24.3%, while the power range for single-tissue methods fell from 23.0%-28.0% to 14.3%-17.7% under the same conditions (Fig 3a). A similar trend was also observed at the gene level (Fig 3b).

In terms of SNP effects on gene expression ($PVE_1$), we observed that the performance of all methods improved with increasing $PVE_1$, and their performance ranking remained stable, with mFABIO consistently outperforming TGFM and both multi-tissue methods surpassing the single-tissue methods, at both gene-tissue pair level (S4a Fig) and gene level

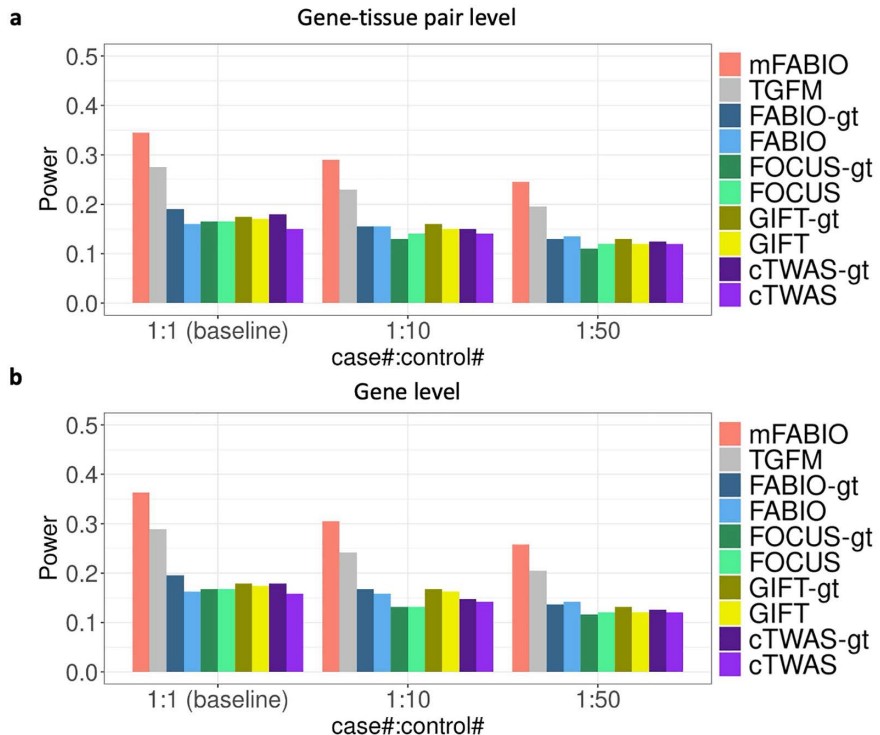

**Fig 2. Power of different methods under the true FDR threshold of 0.05 across different case:control ratios. (a)** Results at gene-tissue pair level. **(b)** Results at gene level.

(S4b Fig). Specifically, for the most challenging setting ($PVE_1$ = 5%), mFABIO still achieved a power of 19.0% at the gene-tissue pair level and 19.5% at the gene level. This was considerably higher than TGFM (15.0% and 15.4%, respectively) and the single-tissue methods (which ranged from 7.5%-10.5% and 7.7%-10.8%, respectively). Additionally, we had a simulation setting fixing $PVE_1$ at 10% and randomizing the number of causal cis-SNPs, which yielded power trends similar to the baseline setting at both gene-tissue pair level (S4a Fig) and gene level (S4b Fig). Next, we investigated how power was affected by the gene's effects on the outcome trait ($PVE_2$). We found that the power of all methods grew as $PVE_2$ increased, with mFABIO consistently remaining the most powerful method. At gene-tissue pair level, as $PVE_2$ was raised from 0.2% to 0.4% and 0.6%, mFABIO's power rose from 25.5% to 34.5% and 46.5%, respectively. Under the same conditions, TGFM's power increased from 20.5% to 27.5% and 37.5%. The power range for single-tissue methods also improved, increasing from 11.0%-14.0% to 15.0%-19.0% and finally to 21.0%-26.0% (S5a Fig). A similar trend was also observed at the gene level (S5b Fig). When horizontal pleiotropy was introduced to the baseline setting, the power of each method decreased slightly due to confounding from the pleiotropic effects. However, mFABIO still outperformed other methods at both the gene-tissue pair level (S5a Fig) and the gene level (S5b Fig).

Finally, we evaluated how statistical power changes with increasing GWAS or eQTL panel sample sizes. At the gene-tissue pair level, when the GWAS sample size increased from 50K to 100K, 150K, 200K, and 250K, the power of mFABIO increased from 34.5% to 45.5%, 49.0%, 55.0%, and 59.0%, respectively. Under the same settings, TGFM's power increased from 27.5% to 37.5%, 41.0%, 45.0%, and 52.5%. The power of single-tissue methods also improved, rising from 15.0%-19.0% at baseline to 27.0%-31.0% when the GWAS sample size reached 250K (S6a Fig). A similar trend was observed at the gene level (S6b Fig). Increasing the eQTL reference panel size also led to improved power. At the gene-tissue pair level, when the eQTL sample size increased from 500 to 1K, 5K, and 10K, the power of mFABIO

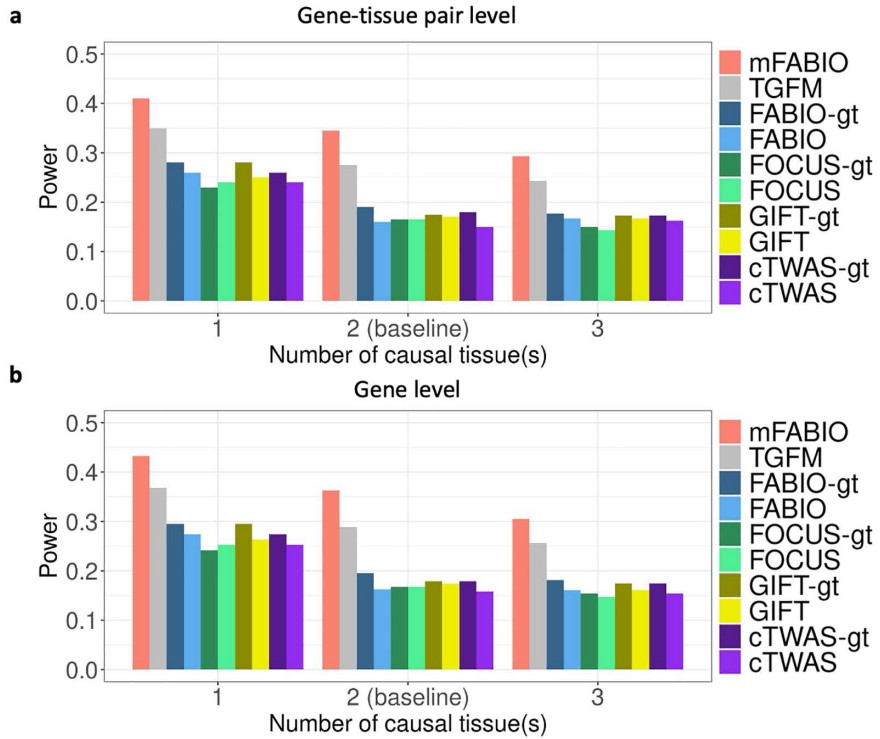

**a** Gene-tissue pair level

**b** Gene level

**Fig 3. Power of different methods under the true FDR threshold of 0.05 across different numbers of causal tissue(s). (a)** Results at gene-tissue pair level. **(b)** Results at gene level.

increased from 34.5% to 40.5%, 57.5%, and 66.5%, respectively, and remained higher than that of all other methods (S7a Fig). The same ranking was observed at the gene level (S7b Fig). Overall, statistical power increased with larger GWAS and eQTL panel sizes, with mFABIO consistently achieving the highest power across scenarios.

The power comparisons described above were based on a true FDR threshold of 0.05, a metric that can only be calculated in simulations. To better mimic a real-data scenario, we also compared power using an estimated FDR threshold of 0.05. We acknowledge this comparison is inherently biased: as shown above, this threshold is well-calibrated for multi-tissue methods but leads to significant FDR inflation for single-tissue methods (S2 and S3 Figs), potentially boosting their apparent power. Despite this unfair advantage, the single-tissue methods did not outperform mFABIO. At the gene-tissue pair level, mFABIO retained the highest power, achieving 34.5% in the baseline setting and averaging 40.0% across all 18 alternative simulations. In comparison, TGFM's power was 28.5% (baseline) and 33.3% (average), while the single-tissue methods' power had a range of 22.0%-26.0% (baseline) and 24.0%-27.0% (average). Though the power of single-tissue methods appeared more comparable under this threshold, it came at the cost of excessive false discoveries. The conclusion holds regardless of the case:control ratio (S8a Fig), the number of causal tissue(s) (S8b Fig), the SNP effect on gene expression (S8c Fig), the number of causal cis-SNPs (S8c Fig), and the gene's effects on the outcome trait (S8d Fig), as well as at the gene level (S9a–S9d Fig). The same conclusion also holds when varying the sample sizes of the GWAS and eQTL panels at both the gene-tissue pair level (S8e and S8f Fig) and the gene level (S9e and S9f Fig). For the single-tissue methods, we also observed that a joint analysis of all gene-tissue pairs by naturally extending their designs was marginally more powerful, than applying them on each tissue and combining the results across tissues. The joint analysis yielded higher average power at both the gene-tissue pair level (24.8%-27.0% vs. 24.0%-24.4%) and the

gene level (25.5%-27.5% vs. 24.4%-25.0%). Therefore, we exclusively used this more powerful joint analysis strategy for all single-tissue methods in our subsequent real-data applications.

**Application to analyzing binary traits in UKBB**

We performed multi-tissue TWAS fine-mapping analysis on six binary disease traits through integrating the GTEx cis-predicted expression models of 38 meta-tissues and the GWAS of UK Biobank (details in Materials and Methods). Specifically, we focused on 13,700 genes and 119,270 unique gene-tissue pairs and examined their associations with each of the six binary traits. The six binary traits include asthma (AS), breast cancer (BRCA), gout (GO), hypertension (HT), prostate cancer (PRCA), and rheumatoid arthritis (RA).

While we did not include age, sex, and principal components as covariates in our main analysis, we assess robustness of such approach by applying mFABIO to the same UK Biobank asthma dataset with and without sex and the top 10 genotype principal components as covariates. The results showed minimal differences between the two sets of analysis: 57 vs. 58 identified genes and 41 vs. 41 identified gene-tissue pairs. For other GWAS summary-statistics-based methods, these covariates had already been adjusted during the generation of GWAS summary statistics, as described in the Materials and Methods.

We first evaluated the results at gene-level among these traits. We identified 1,427 (1,355 unique) GWAS risk genes that contain at least one genome-wide significant SNP ($p < 5 \times 10^{-8}$) across the six disease traits. We performed the TWAS marginal analysis using FUSION for each pair of meta-tissue and disease trait, and considered those genes that are significant in at least one pair of meta-tissue and trait as TWAS risk genes. We identified a total of 479 (470 unique) TWAS risk genes across the six traits. Overlapping the results from the two analyses leads to a total of 1,766 (1,661 unique) risk genes that contain at least one genome-wide significant SNP or shows marginal TWAS significance (Table 1). We consider these risk genes more likely to be the potentially "true" causal genes compared to the rest.

Same as what we did in the simulations, we estimated FDR using the test statistics generated by different methods and declared significant gene associations based on an estimated FDR threshold of 0.05 for all methods. mFABIO identified an average of 42 causal genes per disease, with 60.9% of those genes supported by GWAS or TWAS evidence (Table 1). As a comparison, the other multi-tissue method TGFM, identified an average of 29 causal genes per disease, with 52.2%

**Table 1. Gene-level results of multi-tissue TWAS fine-mapping in UK Biobank.**

| Trait | GWAS risk genes | TWAS risk genes | Risk genes with GWAS/TWAS evidence | mFABIO | TGFM | FABIO | FOCUS | GIFT | cTWAS |
|---|---|---|---|---|---|---|---|---|---|
| AS | 166 | 56 | 195 | 58 (31) | 37 (16) | 108 (35) | 155 (30) | 98 (28) | 92 (18) |
| BRCA | 54 | 13 | 63 | 43 (19) | 29 (10) | 65 (8) | 75 (10) | 54 (9) | 61 (7) |
| GO | 267 | 64 | 296 | 38 (29) | 30 (21) | 72 (26) | 85 (22) | 87 (25) | 75 (20) |
| HT | 769 | 307 | 1,012 | 78 (50) | 55 (30) | 209 (52) | 230 (41) | 203 (49) | 212 (40) |
| PRCA | 157 | 28 | 176 | 21 (17) | 15 (11) | 55 (10) | 85 (10) | 57 (10) | 62 (11) |
| RA | 14 | 11 | 24 | 15 (8) | 12 (5) | 22 (3) | 35 (4) | 20 (3) | 25 (4) |
| Total | 1,427 | 479 | 1,766 | 253 (154) | 178 (93) | 531 (134) | 665 (117) | 519 (124) | 527 (100) |

Table 1 summarizes the number of discoveries for each of the six disease traits (rows) in the multi-tissue TWAS fine-mapping analysis of UK Biobank. A GWAS risk gene (2nd column) is defined as a gene that harbors at least one genome-wide significant SNP (p-value $< 5 \times 10^{-8}$) within 500 Kb of its transcription start site (TSS). A TWAS risk gene (3rd column) is defined as a gene with a marginal significant TWAS p-value after the Bonferroni correction in at least one of the meta-tissue-trait pairs. A risk gene with GWAS/TWAS evidence (4th column) is defined as a gene that harbors at least one genome-wide significant SNP or shows marginal TWAS significance. The last six columns list the number of genes discovered by each of the six methods. The number in the bracket is the number of identified genes that are risk genes with GWAS/TWAS evidence. We used an estimated FDR threshold of 0.05 to declare significance for all methods in the fine-mapping analysis.

of those genes supported by GWAS or TWAS evidence. Among the four single-tissue methods - FABIO, FOCUS, GIFT, and cTWAS - the average numbers of identified causal genes were 88, 110, 86, and 87, respectively, with corresponding proportions of GWAS/TWAS-supported genes at 25.2%, 17.6%, 23.9%, and 19.0% (Table 1). Although mFABIO and TGFM identified fewer causal gene candidates at an FDR of 0.05 compared to single-tissue methods, the proportions of GWAS/TWAS-supported genes were substantially higher. When considering the total number of GWAS/TWAS-supported genes identified across the six disease traits, mFABIO identified at least 14.9% more than any of the other methods (Table 1). Overall, mFABIO not only outperformed other methods in identifying a greater number of GWAS/TWAS-supported gene candidates, but also achieved the smallest candidate gene set with the highest proportion of supported genes compared to those single-tissue methods, both of which highlighted its superior performance at gene-level.

We then evaluated the identified gene-tissue pairs across different methods. In total, we identified 11,613 (11,076 unique) GWAS risk gene-tissue pairs, where the gene of each pair contains at least one genome-wide significant SNP ($p < 5 \times 10^{-8}$) associated with the corresponding trait. Additionally, we identified 2,152 (2,116 unique) TWAS risk gene-tissue pairs that showed significance in our marginal TWAS analysis for each meta-tissue and disease trait pair. Combining these two sets resulted in 13,148 (12,466 unique) risk gene-tissue pairs that either contain a genome-wide significant SNP or show marginal TWAS significance (Table 2). We consider these risk gene-tissue pairs to be more likely representative of the "true" associations with the corresponding disease traits.

At the gene-tissue pair-level, mFABIO identified an average of 65 causal gene-tissue pairs per disease, with 77.2% of those pairs supported by GWAS or TWAS evidence (Table 2). TGFM identified an average of 38 causal pairs per disease, with 66.7% of those pairs supported by GWAS or TWAS evidence. Among the four single-tissue methods - FABIO, FOCUS, GIFT, and cTWAS - the average numbers of identified causal pairs were 145, 167, 132, and 137, respectively, with corresponding proportions of GWAS/TWAS-supported pairs at 30.1%, 24.9%, 30.3%, and 29.3% (Table 2). Similar to our observations at gene-level, mFABIO and TGFM identified fewer causal gene-tissue pairs at an FDR of 0.05 compared to single-tissue methods, however, the proportions of GWAS/TWAS-supported pairs were substantially higher. Again, if we consider the total number of GWAS/TWAS-supported pairs identified across the six disease traits, mFABIO identified at least 14.8% more than any of the other methods (Table 2). Therefore, mFABIO had superior performance at the gene-tissue pair-level as well.

We further evaluated whether significant gene-tissue pairs (estimated FDR < 0.05) were enriched in the likely disease-relevant tissues based on PubMed literature (details in Materials and Methods). We first compared the absolute number of significant pairs found within these tissues. By this metric, mFABIO identified the most relevant pairs for all diseases except hypertension (Fig 4a). The exception for hypertension is likely due to that, single-tissue methods are confounded by a much larger total number of discoveries, which may include a high rate of false positives as shown in Table 2. To account for this, we calculated the proportion of a method's significant pairs located in relevant tissues as a more robust metric. This analysis showed that mFABIO achieved the highest proportion of relevant discoveries for every disease (Fig 4b), with an overall average of 29.7%. This average was higher than that of TGFM (27.2%) and substantially higher than the range for single-tissue methods (12.5%–15.5%). These results demonstrate mFABIO's superior ability to prioritize biologically relevant gene-tissue associations in real-data applications.

We list a few gene-tissue examples that are uniquely identified by mFABIO. The first example is *D2HGDH* (D-2-hydroxyglutarate dehydrogenase) in lung tissue associated with asthma (gene-tissue PIP = 1, marginal TWAS p-value = $2.94 \times 10^{-6}$). *D2HGDH* has been associated with asthma in a previous GWAS [31] through its cis-SNP rs34290285, which is consistent with our GWAS results on asthma that highlighted the same marker within the gene locus (Fig 5a). And the previous lung eQTL analysis of the top SNPs revealed that the G allele of rs34290285 is associated with reduced expression of *D2HGDH* mRNA in the lung [31]. This downregulation leads to the accumulation of the metabolite D-2-hydroxyglutarate (D-2HG). Critically, in murine models, accumulating D-2HG was shown to potently suppress key hallmarks of allergic asthma, including airway hyperreactivity, Th2 cell responses, and IgE-mediated inflammation [32]. This provides a direct mechanistic pathway connecting lower *D2HGDH* expression in the lung to a protective effect against asthma.

**Table 2. Gene-tissue pair-level results of multi-tissue TWAS fine-mapping in UK Biobank.**

| Trait | GWAS risk pairs | TWAS risk pairs | Risk pairs with GWAS/TWAS evidence | mFABIO | TGFM | FABIO | FOCUS | GIFT | cTWAS |
|---|---|---|---|---|---|---|---|---|---|
| AS | 1,333 | 293 | 1,457 | 41 (29) | 33 (19) | 86 (25) | 111 (30) | 88 (26) | 78 (25) |
| BRCA | 372 | 29 | 395 | 40 (33) | 28 (20) | 52 (15) | 58 (15) | 45 (18) | 50 (21) |
| GO | 2,232 | 240 | 2,305 | 95 (74) | 45 (32) | 181 (77) | 215 (68) | 155 (62) | 175 (70) |
| HT | 6,212 | 1,482 | 7,444 | 152 (120) | 84 (58) | 423 (126) | 470 (117) | 385 (116) | 391 (106) |
| PRCA | 1,411 | 88 | 1,475 | 37 (33) | 20 (15) | 95 (14) | 101 (12) | 87 (12) | 89 (14) |
| RA | 53 | 20 | 72 | 26 (13) | 18 (8) | 36 (6) | 51 (8) | 33 (6) | 42 (6) |
| Total | 11,613 | 2,152 | 13,148 | 391 (302) | 228 (152) | 873 (263) | 1,006 (250) | 793 (240) | 825 (242) |

Table 2 summarizes the number of identified gene-tissue pairs for each of the six disease traits (rows) in the multi-tissue TWAS fine-mapping analysis of UK Biobank. A GWAS risk gene-tissue pair (2nd column) is defined as its gene that harbors at least one genome-wide significant SNP (p-value < 5 × 10$^{-8}$) within 500 Kb of its transcription start site (TSS). A TWAS risk gene-tissue pair (3rd column) is defined as a pair with a significant marginal TWAS p-value after the Bonferroni correction in the corresponding meta-tissue-trait analysis. A risk gene-tissue pair with GWAS/TWAS evidence (4th column) is defined as its gene that harbors at least one genome-wide significant SNP or this pair shows marginal TWAS significance. The last six columns list the number of gene-tissue pairs discovered by each of the six methods. The number in the bracket is the number of identified gene-tissue pairs that are risk pairs with GWAS/TWAS evidence. We used an estimated FDR threshold of 0.05 to declare significance for all methods in the fine-mapping analysis.

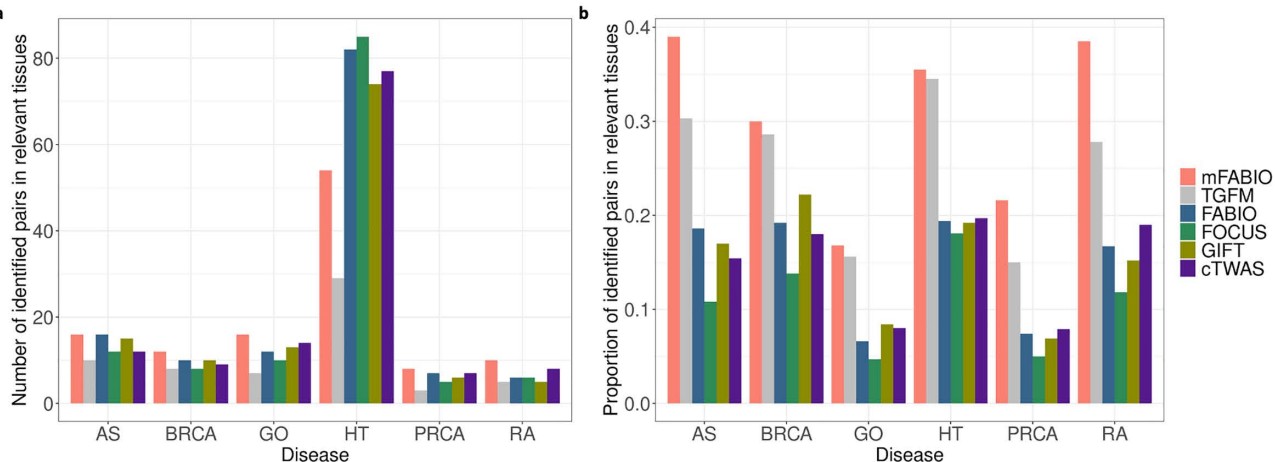

**Fig 4. Comparison of significant gene-tissue pairs located in relevant tissues. (a)** Absolute number of significant gene-tissue pairs located in relevant tissues determined by PubMed literature for each method across diseases. **(b)** Proportion of a method's significant gene-tissue pairs located in relevant tissues determined by PubMed literature across diseases.

The second example is *CYBRD1* (cytochrome b reductase 1) in breast mammary tissue associated with breast cancer (gene-tissue PIP = 1). Notably, while the top cis-SNP in its coding region (rs13020413) did not reach genome-wide significance in previous studies and our GWAS (Fig 5b), *CYBRD1* was flagged as a significant gene in our TWAS pairing the breast mammary tissue with breast cancer (marginal TWAS p-value = 8.90 × 10$^{-6}$). This finding is biologically relevant, as *CYBRD1* (also known as *DCYTB*) plays a crucial role in iron metabolism [33], a process linked to tumor-infiltrating lymphocytes (TILs) and regulated by hypoxia-inducible factor (HIF) in malignant breast cells [34]. Furthermore, *CYBRD1* serves as a prognostic marker for breast cancer, as its increased expression is associated with a favorable prognosis and is implicated in regulating cancer cell proliferation and apoptosis [35]. The successful identification of *CYBRD1*

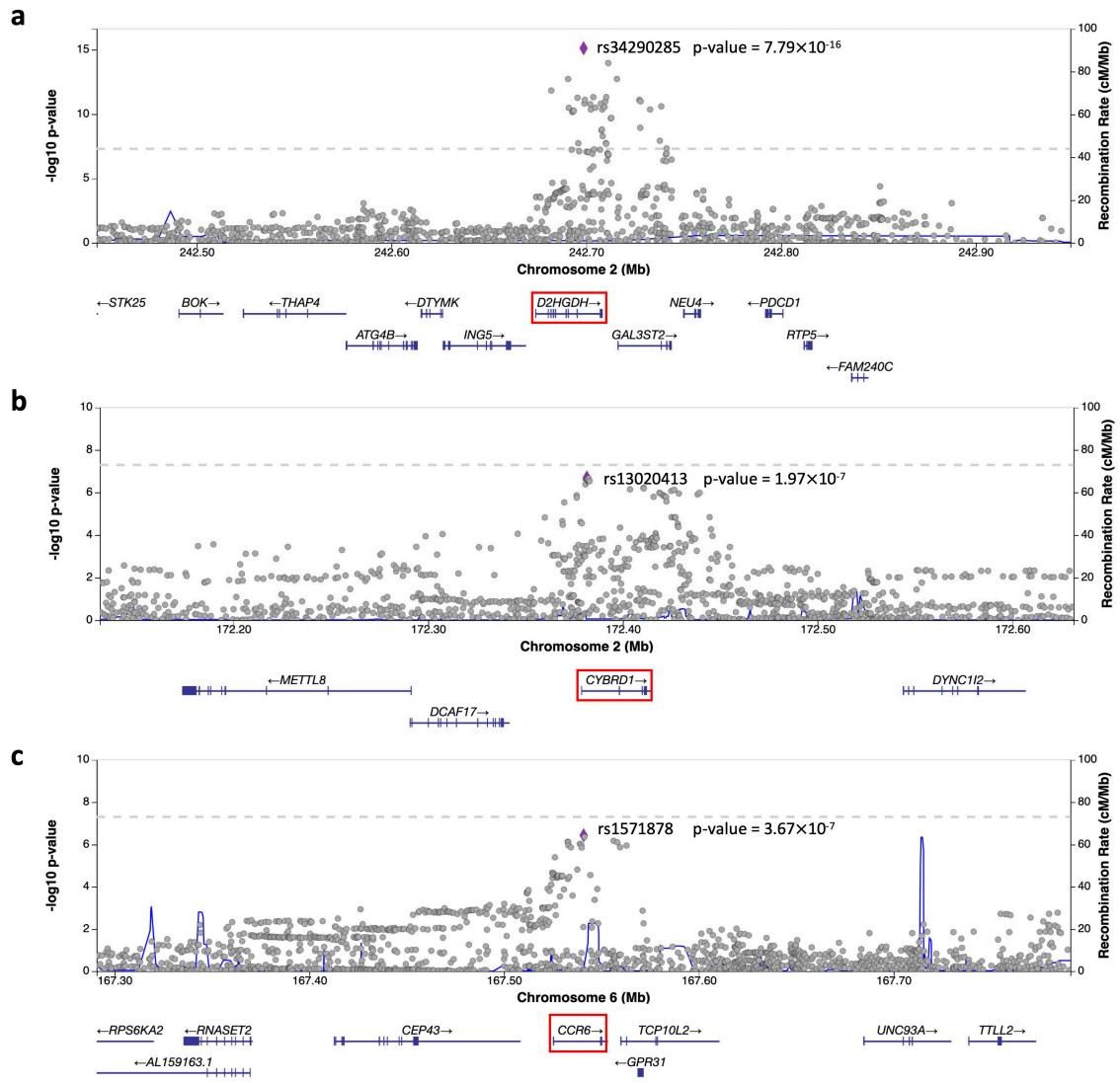

**Fig 5. LocusZoom plots of selected gene candidates uniquely identified by mFABIO. (a)** Plot of the corresponding locus with the GWAS p-value of the top cis-SNP rs34290285 in *D2HGDH* for asthma. **(b)** Plot of the corresponding locus with the GWAS p-value of the top cis-SNP rs13020413 in *CYBRD1* for breast cancer. **(c)** Plot of the corresponding locus with the GWAS p-value of the top cis-SNP rs1571878 in *CCR6* for rheumatoid arthritis. All these LocusZoom plots were generated using [39].

demonstrates that mFABIO can nominate credible, tissue-specific candidate genes that lack strong statistical support from traditional GWAS.

The third example is *CCR6* (C-C chemokine receptor type 6) in spleen associated with rheumatoid arthritis (gene-tissue PIP = 0.98). While the top cis-SNP for *CCR6* in our GWAS (rs1571878) was not genome-wide significant (Fig 5c), it has been previously associated with rheumatoid arthritis (RA) in another GWAS study [36]. Also, *CCR6* is a significant gene candidate in our TWAS pairing spleen with RA (marginal TWAS p-value = $3.92 \times 10^{-6}$). The involvement of the spleen in RA is well-established, as spleen is a major hub for immune activity, and its enlargement is a characteristic of Felty's syndrome, a known complication of severe RA [37]. This connection is further strengthened by mechanistic evidence from animal models. One study found that in mice with collagen-induced arthritis, regulatory T-cells (Tregs) in the spleen

significantly increased their expression of *CCR6* compared to the healthy control [38]. This suggests that during arthritic inflammation, splenic Tregs upregulate *CCR6*, a key receptor that mediates the migration of immune cells to inflammatory sites.

## Benchmark of runtime and memory usage

Although mFABIO employs a relatively complex Bayesian inference procedure with a probit link designed for binary outcomes, we observed competitive computational efficiency compared with other methods. In simulations, at the largest simulated GWAS sample size ($N = 250,000$), mFABIO required approximately 2.66 hours and 8 GB of memory to sequentially fine-map all 50 LD loci in a single simulation replicate. In comparison, summary-statistics-based methods required 1.95-3.78 hours in total, including approximately 1.7 hours to compute GWAS summary statistics and 0.25-2.08 hours for the fine-mapping step. Memory usage for these methods was about 8 GB for GWAS summary statistics computation and 3–6 GB for the subsequent fine-mapping stage.

In real data analysis using 337,198 individuals from the UK Biobank, and utilizing 8 CPU threads, mFABIO required an average of 9.77 hours to sequentially fine-map all LD loci on a single chromosome (with an average of 1,430 cis-SNPs per LD locus, and 65 LD loci per chromosome). By comparison, methods relying on GWAS summary statistics required an average of 7.21 hours per chromosome to generate the necessary GWAS summary statistics, followed by 1.02 hours for fine-mapping with TGFM and 0.90-5.84 hours for single-tissue methods. Regarding memory usage, the peak memory requirement for mFABIO was approximately 10 GB. In contrast, other methods required approximately 16 GB of peak memory during the GWAS summary statistics precomputation step, in addition to 4–8 GB of memory for the subsequent fine-mapping stage.

## Discussion

In this study, we introduce mFABIO, a novel multi-tissue TWAS fine-mapping method specifically designed for binary outcomes. Unlike existing approaches, mFABIO employs a probit model to appropriately capture the binary nature of disease traits and jointly models expression across genes and tissues, accounting for their GReX correlations. This joint modeling approach enables mFABIO to produce well-calibrated test statistics and achieve greater statistical power than current single-tissue and multi-tissue methods designed for continuous traits. Through extensive simulations and real-world applications to six disease phenotypes from the UK Biobank, we demonstrate the practical advantages of mFABIO.

mFABIO models the relationship between genetically regulated expression (GReX) and binary outcomes using a probit link function. This function introduces a latent liability score, where an individual is classified as having the disease if their score surpasses a certain threshold. A common alternative to the probit link is the logistic link function, which directly models disease probability. However, in practice, both approaches often yield similar results [40]. Our simulations further confirm that mFABIO performs well when the data were simulated using a logistic model, suggesting that mFABIO can effectively capture GReX-outcome relationships, even when its underlying modeling assumptions are not perfectly met.

mFABIO is not without its limitations. First, mFABIO's probit link function requires individual-level GWAS data to infer latent liability scores, which precludes the direct use of more widely available GWAS summary statistics. While requiring individual-level data may limit the application of mFABIO, it helps mitigate biases commonly seen in methods that rely solely on GWAS summary statistics, where LD mismatches between the reference panel and the analyzed GWAS data frequently occur [41,42]. Second, like many other methods compared in this study, mFABIO uses a two-stage TWAS fine-mapping strategy. In the first stage, it uses the eQTL mapping study to construct the gene expression prediction model, while in the second stage, it applies the SNP weights inferred from the eQTL mapping study to construct GReX in the GWAS data for TWAS fine-mapping. Although convenient, this strategy ignores the expression prediction uncertainty introduced in the first stage, which can lead to a loss of statistical power [6,22,43]. A key future direction is to develop a joint-likelihood framework for mFABIO that simultaneously models both stages, accounting for this expression prediction uncertainty.

We note that TWAS fine-mapping methods, including mFABIO, rely on eQTL reference panels to construct gene expression prediction models, and these panels may originate from independent cohorts across tissues. To assess the robustness of mFABIO under this scenario, we performed an additional analysis in which one tissue-specific eQTL panel (GTEx whole blood) was replaced with an independent reference panel (PBMC) from a different study (details in Materials and Methods). In the analysis, we observed that SNP weights from the two panels (GTEx and PBMC) were reasonably well correlated (S10a Fig). In fine-mapping analyses, mFABIO identified the same set of significant genes for blood tissue, and the corresponding PIPs were also highly correlated between panels (S10b Fig). These results suggest that mFABIO remains robust when eQTL reference panels originate from independent cohorts, provided that ancestry is matched and appropriate harmonization procedures are applied. We note, however, that integration becomes more challenging when tissues are derived from cohorts with different genetic ancestries due to differences in LD structure and allele frequencies. In such cases, additional harmonization may be required to construct a unified set of SNP weights prior to fine-mapping. Methods such as SR-TWAS [44] (Stacked Regression TWAS) provide effective approaches for integrating SNP weights across multiple reference panels, although they require access to individual-level eQTL training data.

It is also worth noting that the current mFABIO model includes a polygenic term $X\alpha$ with a Gaussian prior on $\alpha$, which is statistically equivalent to a random-effect component with covariance determined by the SNPs in $X$. Consequently, the model implicitly captures correlations among individuals arising from their cis-genotype similarities within the local genomic region. Nevertheless, incorporating additional random-effect components may further improve model flexibility. For example, random effects derived from trans-genotypes or pedigree information could capture other sources of individual similarity. Extending the framework to a more general mixed-effects model that allows arbitrary covariance structures would be valuable, although computational challenges may require structural assumptions such as low-rank approximations.

Finally, like other methods, mFABIO requires that the gene expression study has the same ancestry as the GWAS data. In our application, the GTEx data, which contains samples of predominantly European-ancestry, aligns well with the UKBB cohort, also primarily of European-ancestry. A key next step is to develop TWAS fine-mapping methods that can leverage and accommodate ancestry diversity, especially as multi-ancestry GWAS data becomes more common. Despite these limitations, mFABIO remains a robust and powerful method for fine-mapping likely causal tissues and genes for binary outcomes.

## Supporting information

**S1 Fig. Number of false signals in null simulations.** (a) The average number of false positive gene-tissue pairs per simulation replicate across null settings using an estimated FDR threshold of 0.05. (b) The average number of false positive genes per simulation replicate across null settings using an estimated FDR threshold of 0.05.
(TIFF)

**S2 Fig. The true false discovery rate (FDR) in alternative simulations at the gene-tissue pair level.** We calculated the true FDR under an estimated FDR threshold of 0.05 to evaluate the calibration of the methods in different simulation settings: (a) different case:control ratios; (b) different numbers of causal tissue(s); (c) different proportions of gene expression variance explained by genetic effects (PVE1) and different numbers of causal cis-SNPs; (d) different proportions of the phenotype's variance explained by causal gene-tissue pairs (PVE2); (e) different GWAS sample sizes; (f) different eQTL sample sizes.
(TIFF)

**S3 Fig. The true false discovery rate (FDR) in alternative simulations at the gene level.** We calculated the true FDR under an estimated FDR threshold of 0.05 to evaluate the calibration of the methods in different simulation settings: (a) different case:control ratios; (b) different numbers of causal tissue(s); (c) different proportions of gene expression variance

explained by genetic effects (PVE1) and different numbers of causal cis-SNPs; (d) different proportions of the phenotype's variance explained by causal gene-tissue pairs (PVE2); (e) different GWAS sample sizes; (f) different eQTL sample sizes. (TIFF)

**S4 Fig. Power of different methods under the true FDR threshold of 0.05 across different PVE1 and numbers of causal cis-SNPs.** (a) Results at gene-tissue pair level. (b) Results at gene level. (TIFF)

**S5 Fig. Power of different methods under the true FDR threshold of 0.05 across different PVE2.** (a) Results at gene-tissue pair level. (b) Results at gene level. (TIFF)

**S6 Fig. Power of different methods under the true FDR threshold of 0.05 across different GWAS sample sizes.** (a) Results at gene-tissue pair level. (b) Results at gene level. (TIFF)

**S7 Fig. Power of different methods under the true FDR threshold of 0.05 across different eQTL sample sizes.** (a) Results at gene-tissue pair level. (b) Results at gene level. (TIFF)

**S8 Fig. Statistical power in alternative simulations at the gene-tissue pair level.** We calculated the power under an estimated FDR threshold of 0.05 to evaluate the performance of the methods in different simulation settings: (a) different case:control ratios; (b) different numbers of causal tissue(s); (c) different proportions of gene expression variance explained by genetic effects (PVE1) and different numbers of causal cis-SNPs; (d) different proportions of the phenotype's variance explained by causal gene-tissue pairs (PVE2); (e) different GWAS sample sizes; (f) different eQTL sample sizes. (TIFF)

**S9 Fig. Statistical power in alternative simulations at the gene level.** We calculated the power under an estimated FDR threshold of 0.05 to evaluate the performance of the methods in different simulation settings: (a) different case:control ratios; (b) different numbers of causal tissue(s); (c) different proportions of gene expression variance explained by genetic effects (PVE1) and different numbers of causal cis-SNPs; (d) different proportions of the phenotype's variance explained by causal gene-tissue pairs (PVE2); (e) different GWAS sample sizes; (f) different eQTL sample sizes. (TIFF)

**S10 Fig. (a) Scatter plot of comparing SNP weights generated from the PBMC eQTL panel against those from the GTEx whole blood eQTL panel.** (b) Scatter plot of comparing gene PIPs of mFABIO using the PBMC eQTL panel against using the GTEx whole blood eQTL panel. (TIFF)

**S1 Text. Supplementary text for the methods.** (DOCX)

## Author contributions

**Data curation:** Haihan Zhang.

**Formal analysis:** Haihan Zhang.

**Funding acquisition:** Lam C. Tsoi, Xiang Zhou.

**Methodology:** Haihan Zhang, Kevin He, Lam C. Tsoi, Xiang Zhou.

**Software:** Haihan Zhang.

**Supervision:** Kevin He, Lam C. Tsoi, Xiang Zhou.

**Validation:** Haihan Zhang.

**Visualization:** Haihan Zhang.

**Writing – original draft:** Haihan Zhang.

**Writing – review & editing:** Haihan Zhang, Kevin He, Lam C. Tsoi, Xiang Zhou.

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
