## [Decision Letter · Decision Letter 0]

21 Oct 2025

PGENETICS-D-25-00986

mFABIO: An integrative multi-tissue TWAS fine-mapping approach to prioritize potentially causal genes and tissues underlying binary traits

PLOS Genetics

Dear Dr. Zhang,

Thank you for submitting your manuscript to PLOS Genetics. After careful consideration, we feel that it has merit but does not fully meet PLOS Genetics's publication criteria as it currently stands. Therefore, we invite you to submit a revised version of the manuscript that addresses the points raised during the review process.

Please submit your revised manuscript within 60 days Dec 20 2025 11:59PM. If you will need more time than this to complete your revisions, please reply to this message or contact the journal office at plosgenetics@plos.org. Please include the following items when submitting your revised manuscript:

We look forward to receiving your revised manuscript.

Kind regards,

Lin S. Chen, Ph.D.

Academic Editor

PLOS Genetics

Michael Epstein

Section Editor

PLOS Genetics

Aimée Dudley

Editor-in-Chief

PLOS Genetics

Anne Goriely

Editor-in-Chief

PLOS Genetics

**Additional Editor Comments:**

Both reviewers find the manuscript methodologically sound, recognizing its potentials in multi-tissue TWAS fine-mapping for binary traits. They commend its improved power but request clarification on handling pleiotropic SNP effects, correlations among GReX, covariate adjustment, computational scalability, and use of heterogeneous eQTL references. Minor revisions to equations, figures, and tables are also suggested for clarity. We hope to see these points being addressed in the revision.

**Journal Requirements:**

At this stage, the following Authors/Authors require contributions: Haihan Zhang, Kevin He, Lam C. Tsoi, and Xiang Zhou. Please ensure that the full contributions of each author are acknowledged in the "Add/Edit/Remove Authors" section of our submission form.

The list of CRediT author contributions may be found here: https://journals.plos.org/plosgenetics/s/authorship#loc-author-contributions

https://journals.plos.org/plosgenetics/s/submission-guidelines#loc-parts-of-a-submission

5) We have noticed that you have uploaded Supporting Information files, but you have not included a list of legends. Please add a full list of legends for your Supporting Information files after the references list.

Potential Copyright Issues:

i) We note that  Figure 1 is created through BioRender. Please confirm that you hold a Premium account and provide a pdf copy of the CC BY 4.0 Licence as provided by BioRender. For instructions on how to generate a CC BY 4.0 license for your figure, please see the guidelines here: https://help.biorender.com/hc/en-gb/articles/21282341238045-Publishing-in-open-access-resources.

If you are using the free assets from BioRender, we are unable to publish these images as they are licenced under a stricter licence than CC BY 4.0. In this case we ask you to remove the BioRender images and replace them with open source alternatives.

See these open source resources you may use to replace images / clip-art:

- https://bioart.niaid.nih.gov/

- https://bioicons.com/

- https://healthicons.org/

- https://scidraw.io/

- https://reactome.org/icon-lib

- https://www.phylopic.org/images

- https://journals.plos.org/plosbiology/article?id=10.1371/journal.pbio.3002395

7) Please amend your detailed Financial Disclosure statement. This is published with the article. It must therefore be completed in full sentences and contain the exact wording you wish to be published.

2) If any authors received a salary from any of your funders, please state which authors and which funders..

**Reviewers' comments:**

Reviewer's Responses to Questions

**Comments to the Authors:**

Reviewer #1: The manuscript presents a multi-tissue TWAS fine-mapping method designed for binary traits, accounting for correlations in genetically regulated expression (GReX) of genes. Simulation studies demonstrate that the method provides substantial power gains while maintaining robust control of false discovery rates. Although the proposed method shows good potential, it could be further improved by more clearly addressing how to account for pleiotropic effects from SNPs. In addition, it seems how to account for the correlation among GReX of genes in a LD region is not clearly described in the fine mapping.

Minor comments.

It would be helpful add page number and line numbers.

In model (1)-(4), do the models include all SNPs in a region under consideration. It would helpful to explain how to account for the correlation among SNPs or how to select SNPs in a region to be included in a model. Is it reasonable to assume the effects of SNPs in a region follow iid as in the models?

It seems the function in model (7) do not depend on observed data G_hat and X. It would be helpful to explain how to use the observed data in the parameter estimation.

In the simulation studies, when generating a binary trait via latent variables, only contribution from genes are considered; how about contribution from SNPs? Is it possible some SNP affect gene expression and also have effects on the latent variables not through the genes.

Reviewer #2: The authors present mFABIO, a multi-tissue TWAS fine-mapping framework specifically developed for binary outcomes. Methodologically, mFABIO (i) models a multi-gene, multi-tissue GReX design within genomic loci, (ii) employs a probit latent-liability model suitable for case-control traits, and (iii) adopts a SuSiE-style sum-of-single-effects prior with a hierarchical Dirichlet structure to estimate posterior inclusion probabilities (PIPs) at both the gene and gene-tissue levels. The use of variational Bayes ensures computational scalability. Through extensive simulations and applications to six binary disease traits from the UK Biobank, the method demonstrates improved calibration and higher power compared with single-tissue fine-mapping tools, as well as performance gains over TGFM, while producing more concise and biologically supported candidate sets. The manuscript is clearly written and technically sound. I have the following comments:

1. Equations (1) and (3) incorporate gene-tissue effect sizes and horizontal pleiotropic effects, thereby capturing SNP influences not mediated through gene expression. Could the authors clarify whether mFABIO can also account for covariates such as age, sex, and principal components? Are these effects explicitly modeled within the probit framework, or should users adjust for them externally before applying mFABIO? In addition, it would be interesting to discuss whether the authors plan to extend mFABIO to a mixed-effects framework in the future, which could model correlations among individuals (e.g., related samples or population structure).

2. In the Simulations section, the authors use 50,000 individuals of European ancestry from the UK Biobank for GWAS data and 500 distinct individuals for eQTL mapping. Given the complexity of the Bayesian inference procedure, readers would benefit from a discussion of runtime and memory usage. A comparison of computational performance and statistical power across varying GWAS sample sizes (e.g., 50K, 100K, 150K, 200K, 250K) and eQTL sample sizes (e.g., 500, 1,000, 5,000, 10,000), alongside existing methods such as TGFM, FABIO, FOCUS, GIFT, and cTWAS, would be informative for understanding the scalability, efficiency, and practical utility of mFABIO for large biobank-scale analyses.

3. Both the simulation study and the UK Biobank application rely on eQTL reference data from a single source (500 individuals for the simulations and GTEx for the real data analysis). In many practical settings, eQTL models across tissues may come from different studies or consortia (e.g., GTEx, CMC, ROSMAP, Braineac, CAGE, eQTLGen) (see Methods of doi: 10.1038/s41467-018-04558-1). It would be helpful if the authors could clarify whether mFABIO remains applicable when eQTL reference panels across tissues originate from independent cohorts. For example, does the method require harmonized LD and allele frequencies across tissues, or is it sufficient to provide SNP prediction weights for each gene-tissue pair? A short discussion on this would improve readers’ understanding of mFABIO’s robustness and generalizability.

Other comments:

In Figure 1, the notation “SNP prediction weights for each of the M genes, which can be obtained from eQTL mapping cohorts across different tissues” may be slightly ambiguous. For example, the same notation (w11, …, w1S1) for gene 1 in both tissue 1 and tissue K, which could give the impression that SNP weights are shared across tissues. Clarifying that these weights are tissue-specific would avoid possible misunderstanding.

In Tables 1 and 2, adding a “Total” row summarizing the overall number of significant gene-level and gene-tissue pair-level associations across the six binary disease traits would help readers interpret the aggregate findings more easily. This would also make clearer the meaning of the 1,427 (1,355 unique) GWAS risk genes and 479 (470 unique) TWAS risk genes described in the main text.

In the paragraph describing the third example, “The third example is CCR6 …”, the gene name should be corrected from GCR6 to CCR6 and italicized.

**Have all data underlying the figures and results presented in the manuscript been provided?**

Reviewer #1: **No:** The manuscript should provide links to the data used

Reviewer #2: Yes

PLOS authors have the option to publish the peer review history of their article (what does this mean?). If published, this will include your full peer review and any attached files.

Reviewer #1: No

Reviewer #2: **Yes:** Xihao Li

**Figure resubmission:**
---

## [Decision Letter · Decision Letter 1]

30 Mar 2026

PGENETICS-D-25-00986R1

mFABIO: An integrative multi-tissue TWAS fine-mapping approach to prioritize potentially causal genes and tissues underlying binary traits

PLOS Genetics

Dear Dr. Zhang,

Thank you for submitting your manuscript to PLOS Genetics. The reviewers are largely satisfied with the revised manuscript, though one reviewer still have minor suggestions and concerns. Therefore, we invite you to submit a revised version of the manuscript that addresses the points raised.

Please submit your revised manuscript within by Apr 29 2026 11:59PM. If you will need more time than this to complete your revisions, please reply to this message or contact the journal office at plosgenetics@plos.org. Please include the following items when submitting your revised manuscript:

We look forward to receiving your revised manuscript.

Kind regards,

Lin S. Chen, Ph.D.

Academic Editor

PLOS Genetics

Michael Epstein

Section Editor

PLOS Genetics

Aimée Dudley

Editor-in-Chief

PLOS Genetics

Anne Goriely

Editor-in-Chief

PLOS Genetics

**Journal Requirements:**

1) We notice that your supplementary figures are uploaded with the file type 'Figure'. Please amend the file type to 'Supporting Information'. Please ensure that each Supporting Information file has a legend listed in the manuscript after the references list.

**Reviewers' comments:**

Reviewer's Responses to Questions

Reviewer #1: In this revision, the authors have been very responsive to the issues raised by the reviewers. I have only minor comments:

1. Page 1, lines 115, it says “These GreX values are assumed to have been pre-computed using standard software such as SuSiE”. Can you double check this is true? My understanding is SuSiE is SNP-based fine mapping method, not for building prediction models for gene expression.

2. When analyzing multiple tissues, gene expression levels (or their genetically predicted values) across tissues may be highly correlated or even identical, leading to a singular covariance matrix. How do you address this issue in such extreme cases?

3. In a single LD region, the number of SNPs analyzed can exceed 1,000. Does this pose any computational challenges?

4. In model (1) or (3), the authors assume both the intercept mu and the residue follow normal distributions. How do they use difference information from the data?

Page 10, line 181: Assuming that the vector γ follows a multinomial distribution, Multinomial(1,π), implies that there is at least one causal gene in the region under consideration. If mFABIO is applied to all regions without pre-screening (e.g., using marginal TWAS to identify candidate regions likely to contain causal genes), it may identify putative causal genes in regions that do not truly harbor any causal genes. Please clarify how mFABIO avoids such false positives.

Reviewer #2: I appreciate the efforts made by the authors in addressing my comments thoroughly, which have further enhanced the manuscript. I would like to recommend this manuscript for publication.

Reviewer #3: This manuscript presents mFABIO, a multi-tissue TWAS fine-mapping framework for binary traits that jointly models gene–tissue effects and accounts for correlations in genetically regulated expression. Using a probit model and a SuSiE-style prior, the method enables efficient inference at both gene and gene-tissue levels. Simulation studies and applications to UK Biobank data demonstrate improved power while maintaining appropriate control of false discoveries compared with existing approaches. Overall, the manuscript is well written and methodologically sound. I recommend this manuscript for publication.

**Have all data underlying the figures and results presented in the manuscript been provided?**

Reviewer #1: Yes

Reviewer #2: Yes

Reviewer #3: Yes

PLOS authors have the option to publish the peer review history of their article (what does this mean?). If published, this will include your full peer review and any attached files.

Reviewer #1: No

Reviewer #2: **Yes:** Xihao Li

Reviewer #3: No

**Figure resubmission:**
---

## [Decision Letter · Decision Letter 2]

4 May 2026

Dear Dr Zhang,

We are pleased to inform you that your manuscript entitled "mFABIO: An integrative multi-tissue TWAS fine-mapping approach to prioritize potentially causal genes and tissues underlying binary traits" has been editorially accepted for publication in PLOS Genetics. Congratulations!

Yours sincerely,

Lin S. Chen, Ph.D.

Academic Editor

PLOS Genetics

Michael Epstein

Section Editor

PLOS Genetics

Aimée Dudley

Editor-in-Chief

PLOS Genetics

Anne Goriely

Editor-in-Chief

PLOS Genetics

BlueSky: @plos.bsky.social

Comments from the reviewers (if applicable):

Reviewer's Responses to Questions

**Comments to the Authors:**

Reviewer #1: The authors have responded well to the issues raised by the reviewer. I have one minor question:

In Model/Equation (1), some covariates C may have fixed effects (e.g., sex); is it appropriate to treat their effects as random?

**Have all data underlying the figures and results presented in the manuscript been provided?**

Reviewer #1: Yes

PLOS authors have the option to publish the peer review history of their article (what does this mean?). If published, this will include your full peer review and any attached files.

Reviewer #1: No

**Data Deposition**

http://datadryad.org/submit?journalID=pgenetics&manu=PGENETICS-D-25-00986R2

**Press Queries**

---

## [Editor Report · Acceptance letter]

PGENETICS-D-25-00986R2

mFABIO: An integrative multi-tissue TWAS fine-mapping approach to prioritize potentially causal genes and tissues underlying binary traits

Dear Dr Zhang,

We are pleased to inform you that your manuscript entitled "mFABIO: An integrative multi-tissue TWAS fine-mapping approach to prioritize potentially causal genes and tissues underlying binary traits" has been formally accepted for publication in PLOS Genetics! Your manuscript is now with our production department and you will be notified of the publication date in due course.

With kind regards,

Anita Estes

PLOS Genetics

On behalf of:
